# CURVED DATA REPRESENTATIONS IN DEEP LEARNING

## ABSTRACT

The phenomenal success of deep neural networks inspire many to understand the inner mechanisms of these models. To this end, several works have been studying geometric properties such as the intrinsic dimension of latent data representations produced by the layers of the network. In this paper, we investigate the *curvature* of data manifolds, i.e., the deviation of the manifold from being flat in its principal directions. We find that state-of-the-art trained convolutional neural networks have a characteristic curvature profile along layers: an initial increase, followed by a long phase of a plateau, and tailed by another increase. In contrast, untrained networks exhibit qualitatively and quantitatively different curvature profiles. We also show that the curvature gap between the last two layers is strongly correlated with the performance of the network. Further, we find that the intrinsic dimension of latent data along the network layers is not necessarily indicative of curvature. Finally, we evaluate the effect of common regularizers such as weight decay and mixup on curvature, and we find that mixup-based methods flatten intermediate layers, whereas the final layers still feature high curvatures. Our results indicate that relatively flat manifolds which transform to highly-curved manifolds toward the last layers generalize well to unseen data.

## 1 INTRODUCTION

Real-world data arising from scientific and engineering problems is often high-dimensional and complex. Using such data for downstream tasks may seem hopeless at first glance. Nevertheless, the widely accepted *manifold hypothesis* (Cayton, 2005) stating that complex high-dimensional data is intrinsically low-dimensional, suggests that not all hope is lost. Indeed, significant efforts in machine learning have been dedicated to developing tools for extracting meaningful low-dimensional features from real-world information (Khalid et al., 2014; Bengio et al., 2013). Particularly successful in several challenging tasks such as classification (Krizhevsky et al., 2017) and recognition (Girshick et al., 2014) are deep learning approaches which manipulate data via nonlinear neural networks. Unfortunately, the inner mechanisms of deep models are not well understood at large.

Motivated by the manifold hypothesis and more generally, manifold learning (Belkin & Niyogi, 2003), several recent approaches proposed to analyze deep models by their latent representations. Essentially, a manifold is a topological space locally similar to an Euclidean domain at each of its points (Lee, 2013). A key property of a manifold is its *intrinsic dimension*, defined as the dimension of the related Euclidean domain. Recent studies estimated the intrinsic dimension (ID) along layers of trained neural networks using neighborhood information (Ansuini et al., 2019) and topological data analysis (Birdal et al., 2021). Remarkably, it has been shown that the ID admits a characteristic "hunchback" profile (Ansuini et al., 2019), i.e., it increases in the first layers and then it decreases progressively. Moreover, the ID was found to be strongly correlated with the network performance.

Still, the intrinsic dimension is only a single measure, providing limited knowledge of the manifold. To consider other properties, the manifold has to be equipped with an additional structure. In this work, we focus on Riemannian manifolds which are differentiable manifolds with an inner product (Lee, 2006). Riemannian manifolds can be described using properties such as angles, distances, and curvatures. For instance, the curvature in two dimensions is the amount by which a surface deviates from being a plane, which is completely flat. Ansuini et al. (2019) conjectured that while the intrinsic dimension decreases with network depth, the underlying manifold is highly curved. Our study confirms the latter conjecture empirically by estimating the *principal curvatures* of latent representations of popular deep convolutional classification models trained on benchmark datasets.

Previously, curvature estimates were used in the analysis of trained deep models to compare between two neural networks (Yu et al., 2018), and to explore the decision boundary profile of classification models (Kaul & Lall, 2019). However, there has not been an extensive and systematic investigation that characterizes the curvature profile of data representations along layers of deep neural networks, similarly to existing studies on the intrinsic dimension. In this paper, we take a step forward towards bridging this gap. To estimate principal curvatures per sample, we compute the eigenvalues of the manifold's Hessian, following the algorithm introduced in (Li, 2018). Our evaluation focuses on convolutional neural network (CNN) architectures such as VGG (Simonyan & Zisserman, 2015) and ResNet (He et al., 2016) and on image classification benchmark datasets such as CIFAR-10 and CIFAR-100 (Krizhevsky et al., 2009). We address the following questions:

- How does curvature vary along the layers of CNNs? Do CNNs learn flat manifolds, or, alternatively, highly-curved data representations? How do common regularizers such as weight decay and mixup affect the curvature profile?

- Do curvature estimates of a trained network are indicative of its performance? Is there an indicator that generalize across different architectures and datasets?

- Is there a correlation between curvature and other geometric properties of the manifold, such as the intrinsic dimension? Can we deduce the curvature behavior along layers using dimensionality estimation tools?

Our results show that learned representations span manifolds whose curvature is mostly fixed with relatively small values (on the order of $1e-1$), except for the output layer of the network where curvature increases significantly (on the order of $1$). Moreover, this curvature profile was shared among several different convolutional architectures when considered as a function of the relative depth of the network. In particular, highly-curved data manifolds at the output layer have been observed in all cases, even in mixup-based models (Zhang et al., 2018) which flatten intermediate manifolds more strongly in comparison to non mixup-based networks. In contrast, untrained models whose weights are randomly initialized presented a different curvature profile, yielding completely flat (i.e., zero curvature) manifolds across the last half of the layers. Further, our analysis suggests that estimates of dimensionality based on principal component analysis or more advanced methods need not reveal the actual characteristics of the curvature profile. Finally and similarly to indicators based on the intrinsic dimension (Ansuini et al., 2019; Birdal et al., 2021), we have found that the curvature gap in the last two layers of the network predict its accuracy in that smaller gaps are associated with inferior performance, and larger gaps are related to more accurate models.

## 2 Related Work

Geometric approaches commonly appear in learning-related tasks. In what follows, we narrow our discussion to manifold-aware learning and manifold-aware analysis works, and we refer the reader to surveys on geometric learning (Shuman et al., 2013; Bronstein et al., 2017).

**Manifold-aware learning.** Exploiting the intrinsic structure of data dates back to at least (Belkin & Niyogi, 2004), where the authors utilize the graph Laplacian to approximate the Laplace–Beltrami operator, which further allows to improve classification tools. More recently, several approaches that use geometric properties of the underlying manifold have been proposed. For instance, the intrinsic dimension (ID) was used to regularize the training of deep models, and it was proven to be effective in comparison to weight decay and dropout regularizers (Zhu et al., 2018), as well as in the context of noisy inputs (Ma et al., 2018b). Another work (Gong et al., 2019) used the low dimension of image manifolds to construct a deep model. Focusing on symmetric manifolds, Jensen et al. (2020) propose a generative Gaussian process model which allows non-Euclidean inference. Similarly, Goldt et al. (2020) suggest a generative model that is amenable to analytic treatment if data is concentrated on a low-dimensional manifold. Other approaches aim for a flat latent manifold by penalizing the metric tensor (Chen et al., 2020), and incorporating neighborhood penalty terms (Lee et al., 2021). Additional approaches modify neural networks to account for metric information (Hoffer & Ailon, 2015; Karaletsos et al., 2016; Gruffaz et al., 2021).A recent work (Chan et al., 2022) showed that mapping distributions of real data, on multiple nonlinear submanifolds can improve robustness against label noise and data corruptions.

**Manifold-aware analysis.** Basri & Jacobs (2017) explore the ability of deep networks to represent data that lies on a low-dimensional manifold. The intrinsic dimension of latent representations was used in Ma et al. (2018a) to characterize adversarial subspaces, and to distinguish between learning styles with clean and noisy labels (Ma et al., 2018b). In (Li et al., 2018), the authors employ random subspace training to approximate the ID, and to relate it to problem difficulty. Further, Pope et al. (2020) found that the ID is correlated with the amount of natural image samples required for learning. Subsequently, (Kienitz et al., 2022) investigated the interplay between entanglement and ID, and their effect on the sample complexity. Birdal et al. (2021) harness the formalism of topological data analysis to estimate the ID, and they show it serves as an indicator for the generalization error. Perhaps closest in spirit to our study is the work (Ansuini et al., 2019) where the ID is estimated on several popular vision deep architectures and benchmarks. Their results show that the intrinsic dimension follows a characteristic hunchback profile, and that the ID is negatively correlated with generalization error. Additionally, the authors speculate that latent representations in the final layer of neural networks is highly curved due to the large gap between the ID and the linear dimension (PC-ID) as measured by principal component analysis (PCA).

Beyond dimensionality, other works considered additional properties of the manifold. In (Tosi et al., 2014; Arvanitidis et al., 2018), the authors compute the Riemannian metric to obtain faithful latent interpolations. (Buchanan et al., 2020) studied how DNNs can separate two curves, representing the data manifolds of two separate classes, on the unit sphere. The geometry in deep models with random weights was studied in (Poole et al., 2016), where the authors find that curvature of decision boundaries flatten with depth, whereas data manifolds of e.g., a circular path increase their curvature along network layers. Similarly, Kaul & Lall (2019) also explore the curvature around the decision boundary, and they identify high curvature in transition regions. In contrast, Fawzi et al. (2018) identifies that the decision boundary is mostly flat near data points.

The curvature of latent representations was estimated in (Brahma et al., 2015) using deep belief networks (Hinton et al., 2006) with Swiss roll data and face images. One of the main conclusions was that the manifold flattens with depth. However, their curvature estimates were based on geodesic distances using the connectivity graph, and thus such estimates may be less reliable in settings of sparse and high-dimensional data manifolds. In contrast to (Brahma et al., 2015), it is shown in (Shao et al., 2018) that manifolds learned with variational autoencoders for image data are almost flat. Yu et al. (2018) uses curvature estimates to compare between two neural networks with respect to their fully connected layers. To stabilize computations, the authors propose to augment the data in the neighborhood of every sample. Overall, curvature characterization of latent representations related to deep convolutional models and benchmark datasets is still missing, and thus we focus the current research on this setting.

## 3  BACKGROUND AND METHOD

Given a dataset (e.g., CIFAR-10) and an architecture (e.g., ResNet18), we train the model on the data, and we collect its latent representations along the layers of the model for the train and test sets. Curvature information is estimated for the latent codes, and we perform our analysis on a single curvature quantity, typically the mean absolute value of principal curvatures (see the discussion in App. A), and on the distribution of principal curvatures. In what follows, we briefly describe the extraction of latent codes and the curvature estimation procedure.

**Data density.** In contrast to the intrinsic dimension which is a global feature of the manifold (for connected manifolds), curvature information is a *local property* (Lee, 2006). Additionally, curvatures are based on *second-order derivatives* of the manifold. Consequently, our investigation makes the implicit assumption that data is sufficiently dense for computing curvatures. However, datasets that frequently appear in machine learning, e.g., CIFAR-10, are high-dimensional and sparse, and thus computing local differentiable quantities on such data is extremely challenging. To improve the local density of image samples, we use the same procedure as in (Yu et al., 2018) to generate artificial new samples by reducing the "noise" levels of the original data. Specifically, given an image $I \in \mathbb{R}^{m \times n \times c}$, we denote by $I_j \in \mathbb{R}^{m \times n}$ the matrix at channel $j$. Let $I_j = U \Sigma V^T$ be its SVD, where $U \in \mathbb{R}^{m \times m}$, $V \in \mathbb{R}^{n \times n}$ and $\Sigma \in \mathbb{R}^{m \times n}$ a rectangular diagonal matrix with singular values $\{\sigma_1, \sigma_1, \cdots, \sigma_r\}$ on the diagonal in descending order such that $r$ is the rank of $I_j$. We define $\Sigma'$ as the result of zeroing a subset of singular values in $\Sigma$, allowing to create a new close image

$I'_j = U\Sigma'V^T$. This process is performed along all three R, G, B layers. In our experiments we zeroed all combinations of the ten smallest singular values, generating $1024$ new images.

**Latent representations.** Given an image $I$ of the data of interest, we generate its neighborhood samples using the procedure above, denoted by $\{I'(i)\}$ for $i = 1, \ldots, 1024$. We pass the original image and its neighborhood through the network, and our curvature analysis is performed separately on every such batch. Importantly, passing the input batch of the image and its neighborhood through the nonlinear transformations of the network yields an approximation of a local patch on the manifold, allowing for robust curvature computations. In practice, we extract the latent representations of a subset of layers, similarly to (Ansuini et al., 2019). For instance, in the experiments with ResNets we use the latent codes after every ResNet block and the average pooling before the output. We note that our analyses includes curvature information in the input layer, even though it is shared across different architectures for the same dataset.

**Curvature estimation.** There are multiple approaches to estimate curvature quantities of data representations, see e.g., (Brahma et al., 2015; Shao et al., 2018). We decided to use the algorithm presented in (Li, 2018) and named Curvature Aware Manifold Learning (CAML) since it is backed by theory and it is relatively efficient. CAML requires the neighborhood of a sample, and an estimate of the unknown intrinsic dimension. The ID is computed using the TwoNN algorithm (Facco et al., 2017) on the original dataset (without augmentation) per layer, similarly to (Ansuini et al., 2019).

Let $Y = \{y_1, y_2, \cdots, y_N\} \subset \mathbb{R}^D$ be the data on which we want to estimate the curvature. We assume that the data lies on a $d$-dimensional manifold $\mathcal{M}$ embedded in $\mathbb{R}^D$ where $d$ is much smaller than $D$, thus, $\mathcal{M}$ can be viewed as a sub-manifold of $\mathbb{R}^D$. The key idea behind CAML is to compute a second-order local approximation of the embedding map $f : \mathbb{R}^d \to \mathbb{R}^D$,

$$y_i = f(x_i) + \epsilon_i, \quad i = 1, \ldots, N, \tag{1}$$

where $X = \{x_1, x_2, \cdots, x_N\} \subset \mathbb{R}^d$ are low-dimensional representations of $Y$, and $\{\epsilon_1, \epsilon_2, \cdots \epsilon_N\}$ are corresponding noises.

In the context of this paper, the embedding map $f$ is the transformation that maps the low-dimensional representation of images to a pixel-wise representation which might hold redundant information

To estimate curvature information at a point $y_i \in Y$, we define its neighborhood via the procedure described above, yielding a set of close points $\{y_{i_1}, \ldots, y_{i_K}\}$ where $K$ is the number of neighbors. We use this set and the point $y_i$ to construct via SVD a local natural orthonormal coordinate frame $\left\{\frac{\partial}{\partial x^1}, \cdots, \frac{\partial}{\partial x^d}, \frac{\partial}{\partial y^1}, \cdots, \frac{\partial}{\partial y^{D-d}}\right\}$, composed of a basis for the tangent space (first $d$ elements), and a basis for the normal space. We denote by $x_i$ and $u_{i_j}$ the projection of $y_i$ and $y_{i_j}$ for $j = 1, \ldots, K$ to the tangent space spanned by $\partial/\partial x^1, \ldots, \partial/\partial x^d$, respectively. Importantly, the neighborhood of $y_i$ must be of rank $r > d$, otherwise, SVD can not encode the normal component at $x_i$, yielding poor approximations of $f$ at $x_i$. Thus, we verify that $\{y_{i_1}, \ldots, y_{i_K}\}$ is of rank $d + 1$ or more.

The map $f$ can then be re-formulated with the previously computed locally natural coordinate frame as $f(x^1, \ldots, x^d) = [x^1, \ldots, x^d, f^1, \ldots, f^{D-d}]$. The second-order Taylor expansion of $f^\alpha$ at $u_{i_j}$ with respect to $x_i$ is given by

$$f^\alpha(u_{i_j}) = f^\alpha(x_i) + (u_{i_j} - x_i)^T \nabla f^\alpha + \frac{1}{2}(u_{i_j} - x_i)^T H^\alpha (u_{i_j} - x_i) + \mathcal{O}(|u_{i_j}|_2^2), \tag{2}$$

where $\alpha = 1, \ldots, D - d$, and $u_{i_j}$ is a point in the neighborhood of $x_i$. The gradient of $f^\alpha$ is denoted by $\nabla f^\alpha$, and $H^\alpha = \left(\frac{\partial^2 f^\alpha}{\partial x^i \partial x^j}\right)$ is its Hessian. Given a neighborhood $\{y_{i_1}, \ldots, y_{i_K}\}$ of $y_i$, and their corresponding tangent representations $\{u_{i_j}\}$, we can use Eq. 2 to form a system of linear equations, as we detail in App. E. The principal curvatures are the eigenvalues of $H^\alpha$, and thus estimating curvature information is reduced to a linear regression problem followed by an eigendecomposition. Each hessian has $d$ eigenvalues, therefore each sample will have $(D - d) \times d$ principal curvatures. Finally, we note that one can potentially also compute the Riemannian curvature tensor using the principal curvatures (Yu et al., 2018). However, the latter tensor has an order of $d^4$ elements, and thus it evaluation requires significant time and memory computational resources. Moreover, as the Riemannian curvature tensor is fully determined by the principal curvatures, we base our analysis

on the eigenvalues of the Hessian. For the purpose of evaluating the curvature of manifolds, we estimate the mean absolute principal curvature (MAPC) which is given by the mean of the absolute values of eigenvalues of the estimated hessian matrices.

The code to compute a differentiable estimation of the principal components, and to reproduce our experiments will become available upon acceptance.

## 4 RESULTS

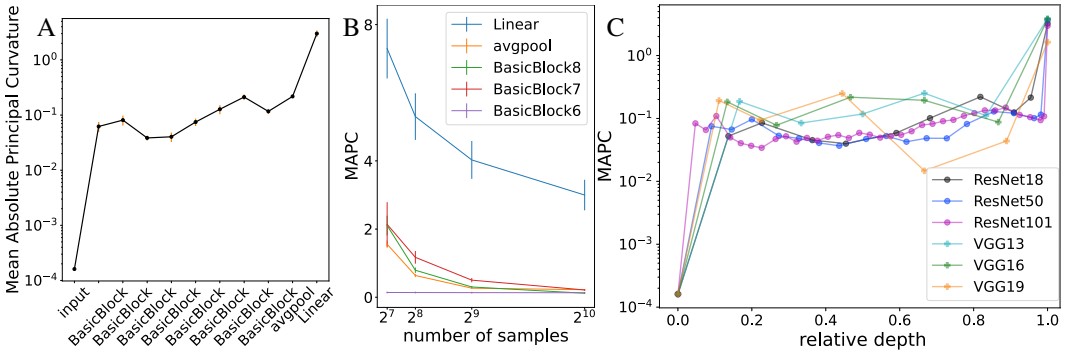

Figure 1: **Mean absolute principal curvature along layers of deep convolutional networks. A)** MAPC and standard deviation as measured for ten seeds using ResNet18 on CIFAR-10 train set. **B)** Repeated evaluation of MAPC on sub-sampled neighborhoods converges for 1024 elements. **C)** MAPC graphs for VGG and ResNet families as a function of the model's relative depth, presenting a characteristic step-like shape in all cases.

### 4.1 DATA MANIFOLDS FEATURE A COMMON CURVATURE PROFILE

We begin our analysis with an empirical evaluation of curvature of latent representations along the layers of a ResNet18 network (He et al., 2016), trained on the CIFAR-10 dataset (Krizhevsky et al., 2009). For a selected subset of layers, we estimate the mean absolute principal curvature (MAPC) as described in Sec. 3. We repeat MAPC evaluation for ten models initialized with random seeds, and we show standard deviation per layer in orange. In Fig. 1 we observe that MAPC is generally *increasing* with depth, demonstrating a large variation of almost four orders of magnitude: MAPC(Input) = 1.6e−4 and MAPC(Linear) = 3.0, see Fig. 1A. Additionally, sharp increases in curvature occur during the transition between the input and the output of the following layer (BasicBlock) as well as between the penultimate to last layers (avgpool to Linear). Notably, MAPC is relatively *fixed* for a majority of network layers.

Our curvature estimates depend directly on the neighborhood around each data point, see Sec. 3. Specifically, sparse and noisy neighborhoods may lead to poor estimates of curvature. We evaluate the robustness of our MAPC computations by evaluating CAML on a repeated sub-sampling of the neighborhood. We observe an overall stable behavior for MAPC values along the last five layers of ResNet18, see Fig. 1B. In particular, MAPC values stabilize in terms of standard deviation when the number of samples per neighborhood reached 1024 elements, and thus utilize a neighborhood of size 1024 in all of our experiments.

To further our exploration on curvature of data manifolds, we investigate whether the characteristic "step-like" shape of MAPC shown in Fig. 1A is shared across multiple networks and datasets. We repeated the above analysis for three variants of a VGG architecture (VGG13, VGG16, VGG19) and three variants of a ResNet architecture (ResNet18, ResNet50, ResNet101) trained on CIFAR-10 and CIFAR-100 datasets, for a total of 12 different models. We show in Fig. 1C six MAPC profiles obtained for CIFAR-10 and plotted with respect to the *relative depth* of the network. Similarly to Raghu et al. (2017), we define the relative depth as the absolute depth of the layer divided by the total number of layers, not counting batch normalizations. Remarkably, the MAPC profiles reveal a common step-like shape, despite the large variation in the underlying models in terms of overall

structure, number of layers, and regularization methods. Beyond their shared behavior, all MAPC graphs attain similar absolute values across network layers, presenting a considerable overlap around the increase in curvature in the last layers. A qualitatively similar plot for CIFAR-10 test set, and CIFAR-100 train set is shown in Fig. 10.

Our results identify that curvature of data manifolds admits a particular trend including three phases: an initial increase, followed by a long phase of a plateau, and ending with an abrupt final increase. These results are consistent with theoretical studies (Cohen et al., 2020), and empirical explorations on neural networks with random weights (Poole et al., 2016). Particularly relevant are the findings in (Ansuini et al., 2019), showing low values of intrinsic dimension (ID) in the last layer of deep convolutional networks, and a large gap between the ID and its linear estimation (PC-ID). The authors propose an indicator for the generalization of the model to unseen data based on the ID values in the last hidden layer, and additionally, they related the gap between PC-ID and ID to the curvature of the data manifold. Motivated by their results and analysis, we suggest a new curvature-based generalization indicator (4.2), and we study the relation between dimensionality and curvature (4.3).

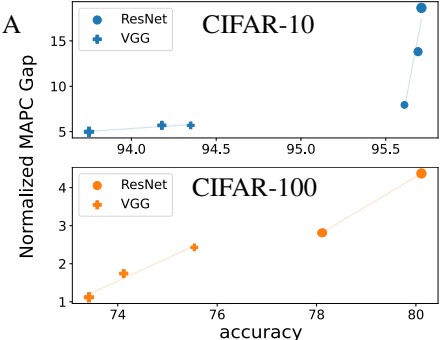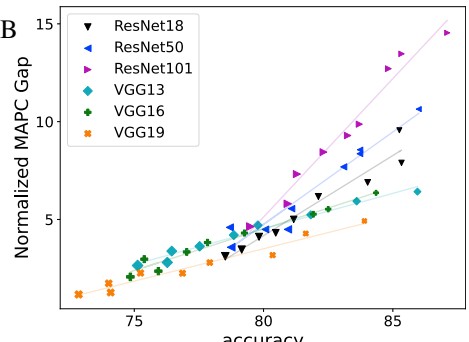

Figure 2: **Normalized MAPC gap is correlated with accuracy. A)** Normalized MAPC gap with respect to model accuracy for six different networks on CIFAR-10 and CIFAR-100. **B)** Normalized MAPC gap with respect to accuracy for six different networks on subsets of CIFAR-100, see text.

## 4.2 CURVATURE GAP IN FINAL LAYERS IS CORRELATED WITH MODEL PERFORMANCE

Our empirical results regarding the curvature profiles for CIFAR-10 and CIFAR-100 (Figs. 1, 10) indicate that MAPC values are higher for CIFAR-10. Moreover, the difference between curvatures in the last two layers of the network, termed MAPC gap from now on, are noticeably smaller for CIFAR-100. In addition, curvature values vary across different models trained on the same dataset. These differences led us to investigate whether the MAPC gap is correlated with the performance of CNNs across architectures and datasets. Specifically, we consider the *normalized MAPC gap* defined as the MAPC gap divided by the average of MAPC across layers, and we compare it against the accuracy of the network. We evaluate the normalized gap on the train sets of CIFAR-10 and CIFAR-100 for the ResNet and VGG families. Each data point corresponds to one of the six models, where the size of the marker represents the network size, e.g., smallest marker for ResNet18 and largest marker for ResNet101 (Fig. 2A). We observe a remarkable correspondence between model performance and the normalized MAPC gap, also emphasized by the additional linear fit graphs per network family. These linear graphs show a consistent trend per family with respect to the difference in gap in relation to difference in accuracy.

To further investigate the correlation between the curvature gap and model performance, we perform the following experiment. We divide the CIFAR-100 dataset which contains a hundred different classes $c_1, c_2, \ldots, c_{100}$ to ten subsets $i \in \{1, \ldots, 10\}$ such that subset $i$ contains samples from classes $c_1$ to $c_{10i}$. We trained all six networks on all subsets, and we computed the normalized MAPC gap and compared it with model performance (Fig. 2B). To improve visibility, we use a different color for every network. Per architecture, each data point corresponds to one of the subsets, where its size represents the size of the subset, e.g., largest markers for the full CIFAR-100 dataset. Similarly to Fig. 2A, we augment the plot with linear fit graphs per architecture. In *all* models and

subsets, we find a remarkable correlation between the normalized MAPC gap and accuracy value. We emphasize that similarly to the ID indicator suggested in (Ansuini et al., 2019), the normalized MAPC gap can be employed without estimating model performance on the test set.

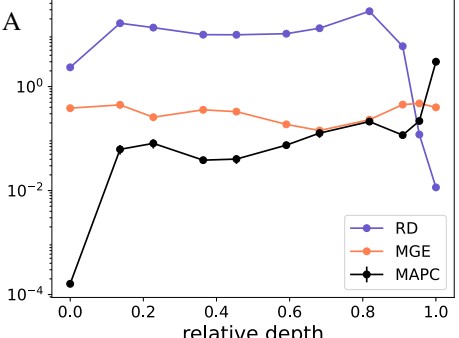 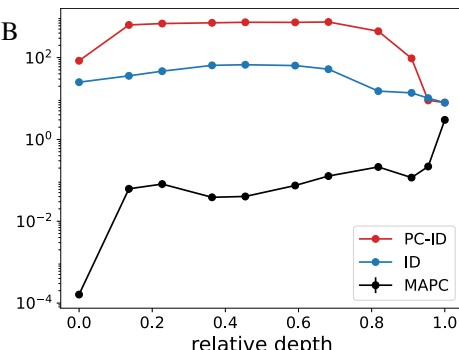

Figure 3: **Comparison of dimensionality and curvature. A)** Relative difference between linear dimension and intrinsic dimension, and maximum gap in eigenvalues of the covariance matrix are compared with MAPC along the relative depth of ResNet18 trained on CIFAR-10. **B)** PC-ID, ID and MAPC for the same network.

### 4.3 DIMENSIONALITY AND CURVATURE OF DATA MANIFOLDS NEED NOT BE CORRELATED

Our third analysis explores the relation between dimensionality and curvature of the data manifold. Existing work on data representations assumes there is a correlation in the flatness of the manifold with respect to dimensionality measures (Verma et al., 2019; Ansuini et al., 2019). On the other hand, analytic examples in geometry such as minimal surfaces (i.e., 2D manifolds) where the principal curvatures are equal and opposite at every point (Do Carmo, 2016), tell us that dimensionality and curvature need not be related. Motivated by these considerations, we ask: how does the dimension correspond to curvature along the network's layers?

To address this question, we extracted the latent representations of a ResNet18 network trained on the CIFAR-10 dataset, and we computed the linear dimension (PC-ID), intrinsic dimension (ID), and mean absolute principal curvature (MAPC). Following Ansuini et al. (2019), PC-ID is defined to be the number of principal components that describe $90\%$ of the variance in the data, and ID is computed using TwoNN (Facco et al., 2017). We focus on the *relative* absolute difference between PC-ID and ID, i.e., $\text{RD} := |\text{PC-ID} - \text{ID}|/\text{ID}$, as a proxy for inferring curvature features, see Fig. 3A. In comparison to the MAPC profile (black), we found no correlation with the relative difference (purple). For instance, RD is high in the first two layers, whereas MAPC is low in the first layer, and then it increases significantly in the second layer. Notably, RD and MAPC admit a weak inverse correlation toward the last three layers of the model. Additionally, we estimate the maximum gap in the eigenvalues of the normalized covariance matrix given by $\text{MGE} := \max_j(\bar{\lambda}_j - \bar{\lambda}_{j+1})$, where $\bar{\lambda}_j$ are the eigenvalues scaled to the range $[0, 1]$. Similarly to the relative difference graph (RD), the maximum gap in eigenvalues (MGE) colored in orange generally does not correspond to MAPC. In particular, MGE in the first and last layers are close in value, whereas MAPC exhibits a difference of four orders of magnitude in those same layers. We also plot PC-ID, ID, and MAPC for the same network in Fig. 3B, allowing to observe the non-relative dimension estimates with respect to MAPC.

### 4.4 UNTRAINED NETWORKS EXHIBIT A DIFFERENT CURVATURE PROFILE

We also computed curvature estimates of data representations along the layers of VGG13, VGG16, and VGG19 for randomly initialized networks. In comparison to MAPC profiles of trained networks (solid lines in 4A), untrained model demonstrate significantly different trends (dashed lines in 4A). While curvature profiles of randomly initialized models and trained networks approximately match up until half of the network depth, there is a sharp decrease in MAPC for untrained models in the second half. Importantly, MAPC values present a similar increase in the first layers for all models, whereas, in the final layers of untrained networks MAPCs are essentially zero. We also note that the decrease in curvature is steeper for larger networks—the orange line (VGG19) is lower than the

green line (VGG16), which in turn, is lower than the cyan line (VGG13), except for the final layer. These results indicate that MAPC profiles of deep convolutional neural networks initially depend on the structure of the model, however, the behavior in the last layers is a direct result of training.

### 4.5 THE EFFECT OF COMMON REGULARIZERS ON CURVATURE

Regularization is a standard practice for modern neural models which are often overparameterized, i.e., the amount of trainable weights is significantly larger than the amount of available training samples (Allen-Zhu et al., 2019). Beyond limiting the parameter space to preferable minimizers, and leading to better generalization properties, certain regularization techniques may affect additional features of the problem. For instance, mixup-based methods which augment training data with convex combinations of the inputs and labels (Zhang et al., 2018) are associated with the flattening of the data manifold (Verma et al., 2019). In their context, flattening means that significant variance directions on the data manifold are reduced. Our curvature estimation framework motivates us to further ask: how do typical regularizers affect curvature statistics of convolutional neural networks?

In the following experiment we investigate this aspect with the baseline model ResNet50 used throughout the paper. This ResNet50 baseline is trained with weight decay of $5e{-}4$ and cosine annealing learning rate scheduling. Additionally, we also train this network with no regularizers, and with manifold mixup and mixup (and no other regularization). We find that all four models demonstrate a step-like profile (Fig. 4B) consistent with our results (Fig.1). In particular, the plateau regime and high final MAPC were observed across all models. Notably, while the networks attained different curvature values in the last layer, the normalized MAPC gap (Fig. 2) distinguishes between the models, and it is correlated with their performance. Namely, we obtain $14, 14, 13, 8$ normalized MAPC gaps for the baseline, manifold mixup, mixup, and no regularization networks, respectively (see their test set accuracy in the legend of Fig. 4B). As per flattening of the data manifold, we note that manifold mixup admits an MAPC profile close in values to our baseline model, whereas mixup shows a significant reduction in curvature (an order of magnitude along most layers in comparison to baseline). Remarkably, mixup does seem to flatten data representations in intermediate layers although it only alters the training samples. In contrast, manifold mixup which manipulates latent codes in a similar fashion to mixup, does not seem to affect MAPC values much. Further, these results reinforce our findings above that high curvature in the last layer, or more precisely, high normalized MAPC gap, is fundamental to the success of the learning model.

### 4.6 DISTRIBUTION OF PRINCIPAL CURVATURES

In our analysis above, we focused on a single estimate of curvature for the entire manifold based on the average absolute value of principal curvatures (i.e., eigenvalues of the Hessian). However, we recall that curvature is a local property for each point of the manifold, and thus curvature variability should also be investigated. Here, we inspect the distribution of principal curvatures for all points at

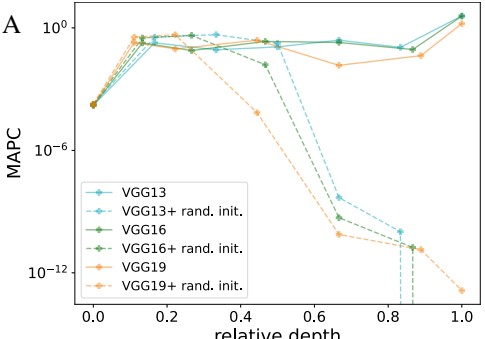 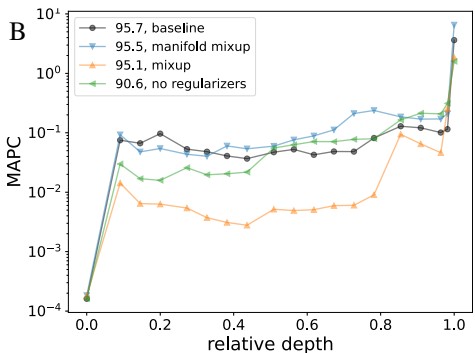

Figure 4: **Comparison of MAPC profiles for baseline models with untrained networks and regularized networks. A)** MAPC graphs for VGG neural architectures before and after training. **B)** MAPC graphs for ResNet50 networks trained with regularizers such as weight decay, learning rate scheduling, and mixup.

every layer. We estimate $(D-d)$ Hessian matrices for each of the 1000 input images for the ResNet family (ResNet18, ResNet50, ResNet101), resulting in $d(D-d)\cdot 1000$ principal curvatures per layer. To analyze this massive amount of information, we compute a histogram per layer, and we plot them overlayed and differentiated by colors according to their relative depth (Fig. 5). For example, light curves are related to the initial layers of the network, whereas dark curves are associated with final network layers. Notably, we observe similar histogram profiles for the majority of intermediate layers (yellow to red curves) across all architectures both in terms of histogram shape and spread of values. Subsequent layers (dark red to brown) present wider distributions, achieving curvature values on the range of $10^2$. Indeed, we observed a mild increase in MAPC toward the last layer of the network (Fig. 1C). The final layer shows that relatively more points attained non-zero curvatures, yielding a histogram profile with a wider base. This result confirms the sharp increase in MAPC of the last layer of convolutional neural networks as shown in (Fig. 1C).

## 5 DISCUSSION

Image classification is a fundamental task which is heavily studied in neuroscience and machine learning. Common wisdom on this problem suggest that *untangling* of manifolds occurs throughout image processing by our vision system and brain (DiCarlo & Cox, 2007), and by deep convolutional neural networks (Bengio et al., 2013). While manifold untangling is commonly perceived as "simpler separability" between class objects (often termed linear separability), defining formal measures of untangling is still an active research topic (Chung et al., 2018). Manifold untangling is typically mentioned alongside *flattening* of the data manifold, a notion related to curvature and to Riemannian geometry. A recent work on this topic distinguishes between the curvature of the decision boundary, and the curvature of the data manifold (Poole et al., 2016), identifying a flattening of the decision boundary with depth and an *opposite* behavior of the data manifold, on deep neural networks with random weights. Additional theoretical and empirical studies provide a mixed picture on this topic, where some works observe flat decision boundaries (Fawzi et al., 2018), and others report highly-curved transition regions (Kaul & Lall, 2019). Further, Brahma et al. (2015) describe the flattening of data manifolds with depth, whereas Shao et al. (2018) essentially observe flat representations. This large variance in results may be attributed to the large variety of different architectures and datasets considered in these works. In this context, our study is the first to investigate systematically how the curvature of latent representations change in common state-of-the-art deep convolutional neural networks used for image classification.

Complementary to existing work on geometric properties of data representations involving their intrinsic dimension (Ansuini et al., 2019; Birdal et al., 2021), and density evolution (Doimo et al., 2020), our study characterizes the *curvature profile* of latent manifolds. The aggregated knowledge arising from prior works on convolutional networks indicate that the intrinsic dimension presents a rapid increase over the first layers, and then it progressively decreases toward the last layers, reaching very low values in comparison to the embedding dimension. In addition, the evolution of the probability density of neighbors as measured for ImageNet (Russakovsky et al., 2015) on several CNN architectures shows almost no overlap with the output and ground-truth distributions

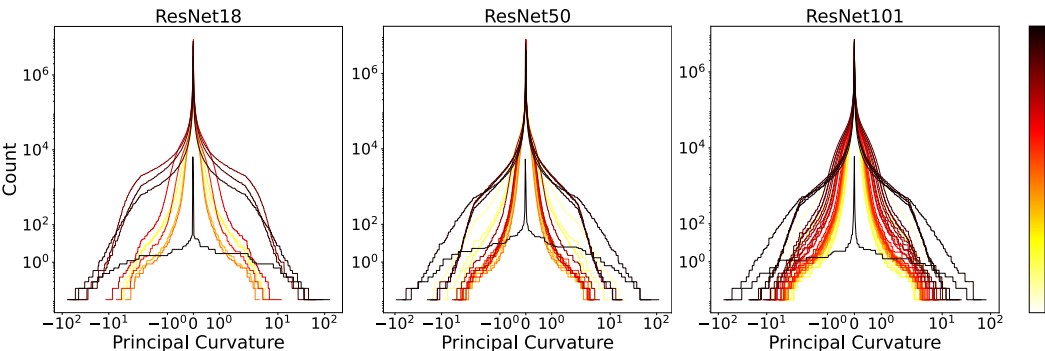

Figure 5: **Distribution of principal curvatures for ResNet models.** Each plot shows the histogram profiles of principal curvatures per layer, colored by their relative depth.

throughout most layers. Specifically, an abrupt overlap emerges in a "nucleation"-type process occurring at layer $142$ of ResNet152 (i.e., toward the final layers of the network). Our exploration adds to this understanding that deep models feature a step-like mean absolute principal curvature profile. For the majority of layers, mean curvature and curvature distribution remain relatively fixed and small in absolute values (Figs. 1, 5). In contrast, a sharp increase in curvature appears in the final layers of the network. Combining our findings with previous work, we obtain a more comprehensive picture of the data manifold: during the first layers, the network maintains almost flat manifolds, allowing samples to move freely between clusters as more directions are available (flat MAPC and high ID). Then, as computation proceeds, samples concentrate near their same-class samples in highly-curved peaks, facilitating separation between clusters. To conclude, we hope that our analysis in this work will inspire others to further our understanding on data manifolds learned with deep neural networks, allowing to develop better and more sophisticated learning models in the future.

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

## A    COMPARING DIFFERENT METRICS OF CURVATURE

The results shown in this paper measure curvature by investigating the Mean Absolute Principal Curvature (MAPC), which is given by the average of the absolute values of eigenvalues of the estimated hessian matrices. To perform a comprehensive analysis, we show the behaviour of three additional metrics that measure curvature. **Mean Absolute Mean Curvature** (MAMC) computes the mean absolute value on the mean curvature, which is the natural extension of mean curvature of surfaces to manifolds in higher dimensions. The mean curvature is defined as the mean principal value, of the hessian matrix. We compute the mean curvature for each one of the $\alpha = 1, \ldots, D - d$ hessian matrices and then take the mean of their absolute values. **Mean Absolute Riemann Curvature** (MARC) computes the mean of the absolute value of all the components in the Riemann curvature tensor. **Mean Absolute Sectional Curvature** computes the mean of the absolute value of the sectional curvatures. As shown in Fig. 6, the pairs MAPC, MAMC and MARC, MASC show a similar trend while MAPC and MASC are larger consistently across different networks. Overall, all the metrics exhibit comparable behaviours and due to the lack of a canonical metric for providing a single scalar value that represents the curvature of a manifold, we opted to use MAPC.

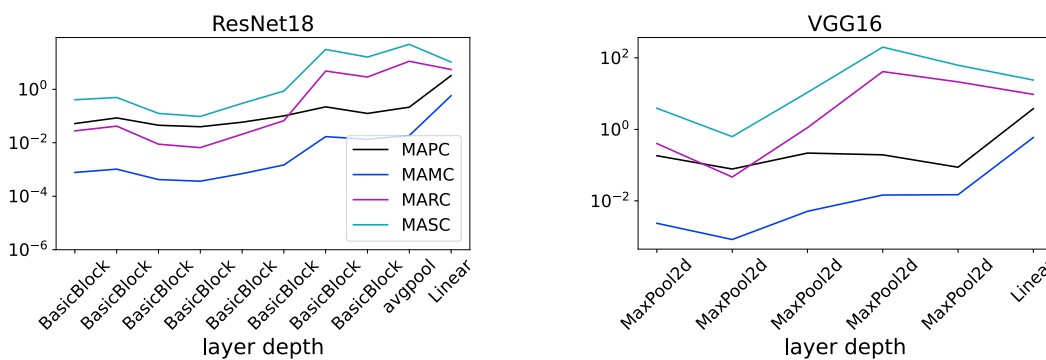

Figure 6: **Comparison of different curvature metrics: MAPC, MAMC, MARC and MASC.**

## B    CURVATURE PROFILE OF UNTRAINED NETWORKS

To augment the results of Sec. 4.4, we added Fig. 7 which shows the MAPC profile for ResNet50 and Resnet101 for untrained networks in comparison to VGG architectures (Fig. 7A) and in comparison to trained ResNet models (Fig. 7B). It is clear that untrained networks exhibit a different MAPC characteristic, most notably by the sharp decrease in the last layer.

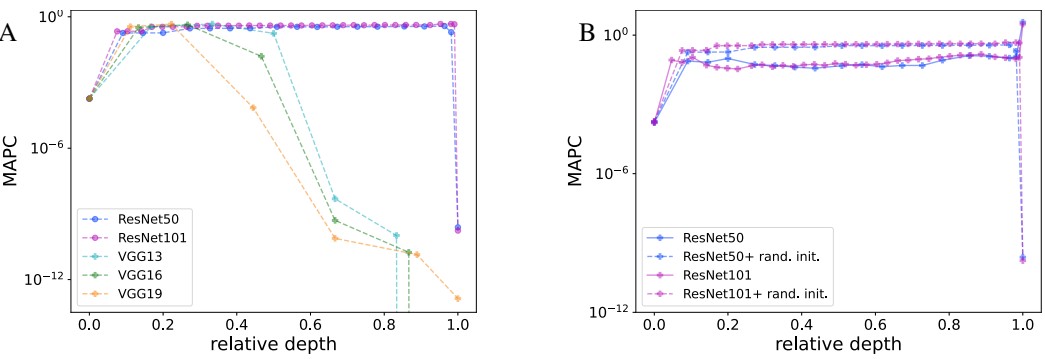

Figure 7: **Comparison of MAPC profiles for baseline models with untrained networks. A)** MAPC graphs for VGG and ResNet neural architectures before training. **B)** MAPC graphs for ResNet50 and ResNet101 networks before and after training.

# C    DATA DENSITY

Curvature estimates for high-dimensional and sparse point clouds are extremely noisy and unreliable. To alleviate this issue, we aimed to (locally) increase the density of the data manifold. Our choice to use SVD is well-motivated from a differential geometry viewpoint. Specifically, the SVD procedure we described in Sec. 3 is closely-related to computing a first order approximation of the manifold at a point, and sampling at the neighborhood of the point. Sampled points may slightly deviate from the data manifold, yet the deviation can be bounded by the absolute value of the modified singular values (which are close to zero in practice). There are several works (Donoho & Grimes, 2003; Zhang & Zha, 2004; Singer & Wu, 2012; Tyagi et al., 2013) that justify the usage of SVD for estimating the tangent plane of a manifold at a given point $p$. In addition to the theoretical justification we provide for the SVD procedure, we also experimented with up-sampling approaches based on standard image transforms, e.g., affine transformations, including translation, shear and rotation (see Fig. 8). That is, we train the networks as before, but during inference, we feed a point with its neighborhood based on image transforms such as rotations of the image. Remarkably, we find for CIFAR10 on ResNet and VGG architectures a characteristic profile akin to the curvature profiles we report e.g., in Fig. 1, with one qualitative different feature. The curvature profiles associated with image transforms exhibit a significant drop in the penultimate layer, whereas curvature profiles as reported in Fig. 1 do not present this characteristic. We believe it is related to the low-dimensionality of the data (as governed by the image transforms) allowing to form a low curvature manifold in comparison to our SVD-based sampling which yields closer points, but with potentially more intrinsic dimensions, making it harder to encode using very low MAPC values. Furthermore, we investigated the proximity of the generated neighborhood using both SVD and affine transformations Fig. 8. The affine transformation included rotations in the range of $[-10, 10]$ degrees, shear parallel to the x and y axis in the range $[-10, 10]$ degrees, horizontal translation in the range $[-imagewidth * 0.1, imagewidth * 0.1]$ and vertical translation in the range $[imageheight * 0.1, imageheight * 0.1]$. Using smaller values for the affine transformation parameters caused the curvature estimation algorithm to fail. It is notable that the generated images using the SVD method create samples that are closer in the sense of euclidean distance along all the layers. Visually, the samples generated using the SVD method look almost identical to the original image from which they were generated Fig. 9.

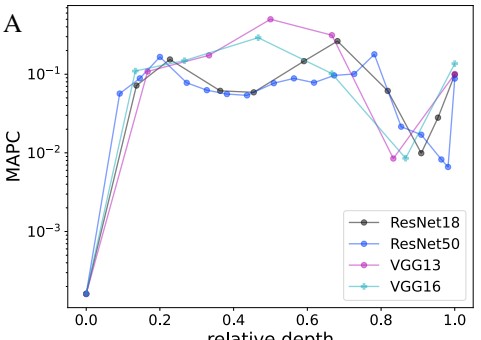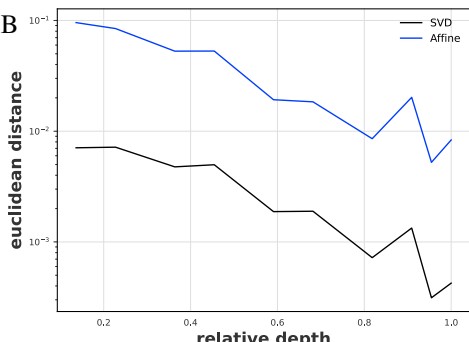

Figure 8: **Effect of generating close samples using affine transformations. A)** MAPC profile estimation using affine transformations for local patch generation. **B)** Comparison of the mean distance between a sample and its neighborhood along the layers of ResNet18.

# D    RELATION WITH PREVIOUS WORK

We provide a more detailed explanation of how our results are consistent with previous work. In (Poole et al., 2016), the authors study how the curvature of the manifold and the decision boundary change along the layers of the network. For data representing a circle, they find that the curvature of the manifold increases with depth, which is consistent with our results. In (Cohen et al., 2020), the authors study the notion of an ensemble of manifolds, where each manifold represents same-class samples, e.g., in binary classification we have two manifolds. Then, the authors define the radius of the ensemble as a normalized value based on the total variance of its anchor points, i.e., points

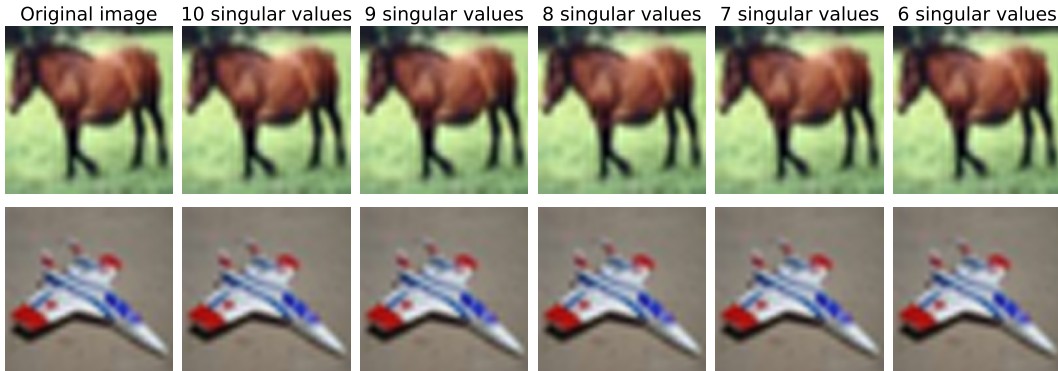

Figure 9: **Visualization of the neighborhood generation process.** The first column to the left shows a sample from the CIFAR10 data set. Each consecutive column shows the generated image using the SVD method where the number of singular values which were set to zero are noted above the column.

which characterize the separation between manifolds. While the radius by the definition of (Cohen et al., 2020) is not a direct measurement of curvature, this measure is related to the geometry of the manifold. Then, the authors compute the radius along the layers of convolutional neural networks such as AlexNet and VGG. In general they find a decrease in the radii and relate it to the classification capacity of CNN. Our work is specifically related to theirs in two aspects: In their Fig. 4, they find that the classification capacity increases significantly in the last layer of CNNs, which may be explained by our empirical findings of increase in curvature. Similarly, in their Fig. 6b they present radius profiles for AlexNet and VGG for smooth 2-d manifolds (generated by image transforms) which are almost a mirror of the MAPC plots we present. Specifically, they show a sharp decrease in the first layer, followed by a plateau from most inner layers, and ending with a further decrease in the last few layers.

## E    ESTIMATING THE HESSIAN MATRIX

As discussed in Sec. 3 above, we wish to estimate the Hessian per embedding mapping $f^\alpha$ where $\alpha = 1, \ldots, D - d$. This is done by building a set of linear equations that solves Eq. 2:

$$f^\alpha(u_{i_j}) = f^\alpha(x_i) + (u_{i_j} - x_i)^T \nabla f^\alpha + \frac{1}{2}(u_{i_j} - x_i)^T H^\alpha (u_{i_j} - x_i) + \mathcal{O}(|u_{i_j}|_2^2),$$

that is $f^\alpha$ is approximated by solving the system $f^\alpha = \Psi X_i$, where $X_i$ contains the unknown elements of the gradient $\nabla f^\alpha$ and the hessian $H^\alpha$. We define $f^\alpha = [f^\alpha(u_{i_1}), \cdots, f^\alpha(u_{i_K})]^T$, where $u_{i_j}$ are points in the neighborhood of $x_i$, projected to the local natural orthogonal frame. The local natural orthonormal coordinate frame is defined as the basis associated with the tangent space and normal space at a point $p$ of the manifold. In practice, the coordinate frame is generated using PCA, where the first $d$ coordinates (associated with the most significant modes, i.e., largest singular values) represent the tangent space, and the rest encode the normal space. Then, we define $\Psi = [\Psi_{i_1}, \cdots, \Psi_{i_K}]$, where $\Psi_{i_j}$ is given via

$$\Psi_{i_j} = \left[u_{i_j}^1, \cdots, u_{i_j}^d, \left(u_{i_j}^1\right)^2, \cdots, \left(u_{i_j}^d\right)^2, \left(u_{i_j}^1 \times u_{i_j}^2\right), \cdots, \left(u_{i_j}^{d-1} \times u_{i_j}^d\right)\right].$$

We solve $f^\alpha = \Psi X_i$ by using the least square estimation resulting in $X_i = \Psi^\dagger f^\alpha$, such that $X_i = \left[\nabla f^{\alpha 1}, \cdots, \nabla f^{\alpha d}, H^{\alpha 1,1}, \cdots, H^{\alpha d,d}, H^{\alpha 1,2}, \cdots, H^{\alpha d-1,d}\right]$, that is, we estimate only the upper triangular part of $H^\alpha$ since it is a symmetric matrix. We do not use the elements of the gradient $\nabla f^\alpha$ for the CAML algorithm, it is only computed as a part of the hessian $H^\alpha$ estimation. We refer the reader for a more comprehensive and detailed discussion in (Li, 2018).

## F    CHARACTERISTIC MEAN ABSOLUTE PRINCIPAL CURVATURE

We complement the results shown in Sec. 4.1, and we demonstrate the mean absolute principal curvature profiles for several networks on CIFAR-10 test set and CIFAR-100 train set as shown in Fig. 10 in the left and right panels, respectively. In both cases we observe the typical behavior described before: an initial sharp increase, followed by a flat phase, and ending with a final increase. Notably, the maximum MAPC values for CIFAR-100 are *lower* in comparison to both CIFAR-10 train and test sets. Moreover, the gap in the final increase in curvature is *smaller* for CIFAR-100. These results are consistent with our discussion in Sec. 4.2.

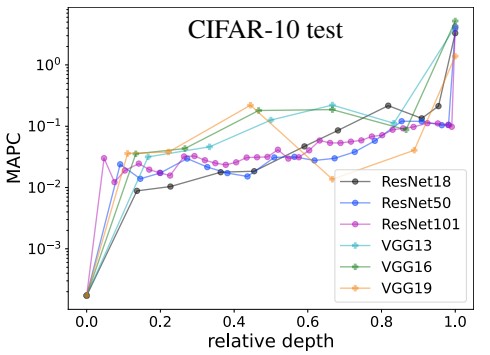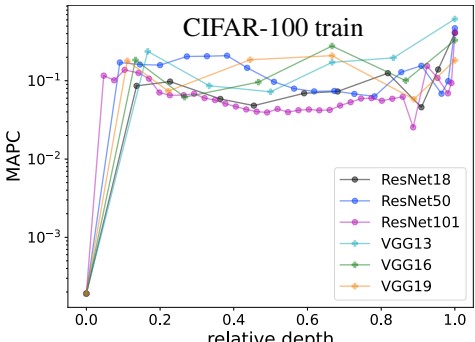

Figure 10: **MAPC profiles on various models and datasets.** MAPC on six different networks on CIFAR-10 test set, left. MAPC on those same models with CIFAR-100 train set, right.

## G    TRAINING DYNAMICS

In Sec. 4, we observed a correspondence between model performance and the normalized MAPC gap. We were interested to see if the training process of the network increases the mentioned gap. We trained a Resnet18 network with CIFAR-10 and observed how the gap changes. We hypothesized that the gap will increase as the network training converges. Remarkably, we indeed find that the normalized MAPC gap is highly correlated with the behavior of the network during training (Fig. 11). Each dot in the plot represents a different snapshot of the model during training, and it is positioned with respect to its accuracy on the test set as a function of the epoch. The points are colored by their normalized MAPC gap (see colorbar on the right). Overall, we observe that during training the accuracy increase in conjunction with the gap, meaning that the network favors a large gap to increase its performance.

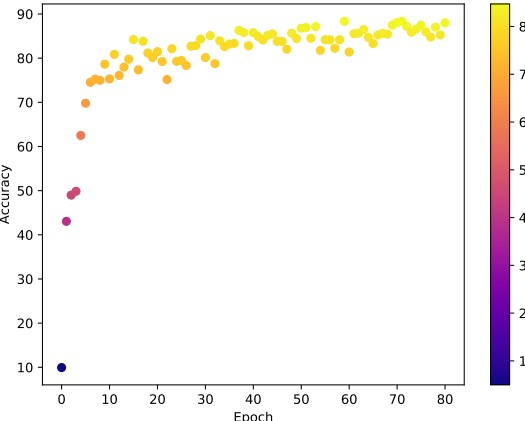

Figure 11: **Training dynamics of the normalized MAPC gap on ResNet18 and CIFAR-10.** The plot shows how the accuracy changes during training, colored by the normalized curvature gap.

## H  RIEMANNIAN GEOMETRY BACKGROUND

This section contains the mathematical background necessary for understanding the curvature estimation process. A vast knowledge in differential geometry is needed to fully comprehend the mathematical background listed below, we will not go in to detail for all the tools we use but rather refer the reader to books on the subject, e.g., (Lee, 2006; Petersen, 2006).

### H.1  PROBLEM STATEMENT

Let $Y = \{y_1, y_2, \cdots, y_N\} \subset \mathbb{R}^D$ be the data on which we want to estimate the curvature. We assume that the data lies on a $d$-dimensional manifold $\mathcal{M}$ embedded in $\mathbb{R}^D$ where $d$ is much smaller than $D$, thus, $\mathcal{M}$ can be viewed as a sub-manifold of $\mathbb{R}^D$. Will will describe how to compute a second-order local approximation of the embedding map $f : \mathbb{R}^d \to \mathbb{R}^D$,

$$y_i = f(x_i) + \epsilon_i , \quad i = 1, \ldots, N , \tag{3}$$

where $X = \{x_1, x_2, \cdots, x_N\} \subset \mathbb{R}^d$ are low-dimensional representations of $Y$, and $\{\epsilon_1, \epsilon_2, \cdots \epsilon_N\}$ are corresponding noises.

### H.2  RIEMANNIAN MANIFOLD

A manifold $\mathcal{M}$ is a topological space that locally resembles Eulidean space near each point. This is particularly useful to our work since the manifold hypothesis states that complex high-dimensional data lies in an intrinsically low-dimensional manifold.

**Definition 1** (Tangent Space). Let $\mathcal{M} \subset \mathbb{R}^D$ be a manifold where $\mathbb{R}^D$ is the ambient space. For every point $\mathbf{p} \in \mathcal{M}$, a tangent space is a vector space that represents the set of all vectors tangent to given differentiable manifold $\mathcal{M}$ at point $p$, denoted by $T_p\mathcal{M}$.

**Definition 2** (Riemannian Manifold). A Riemannian manifold $\langle \mathcal{M}, g \rangle$ is a manifold $\mathcal{M}$ endowed with an inner product $g_\mathbf{p}$ at the tangent space $T_\mathbf{p}\mathcal{M}$ at each point $\mathbf{p}$ that varies smoothly from point to point in the sense that if $X$ and $Y$ are differentiable vector fields on $\mathcal{M}$, then $\mathbf{p} \mapsto g_\mathbf{p}(X(\mathbf{p}), Y(\mathbf{p}))$ is a smooth function.

**Definition 3** (Riemann Curvature (Petersen, 2006)). Let $\langle \mathcal{M}, g \rangle$ be a Riemannian manifold and $\nabla$ the Riemannian connection. The curvature tensor is a $(1, 3)-$ tensor defined by

$$\mathcal{R}(X, Y)Z = \nabla_X \nabla_Y Z - \nabla_Y \nabla_X Z - \nabla_{[X,Y]}Z ,$$

on vector fields $X, Y, Z$. Using Riemannian metric $g$, $\mathcal{R}(X, Y)Z$ can be changed to a $(0, 4)$-tensor:

$$\mathcal{R}(X, Y, Z, W) = g(\mathcal{R}(X, Y)Z, W) .$$

**Definition 4** (Sectional Curvature (Petersen, 2006)). Let $\langle \mathcal{M}, g \rangle$ be a Riemannian manifold, $p \in \mathcal{M}, u, v \in T_p\mathcal{M}$ are two linearly independent tangent vectors, the sectional curvature of the plane $\mathbb{R}u + \mathbb{R}v$ will be defined as

$$K(u, v) = \frac{\mathcal{R}(u, v, u, v)}{\langle u, u \rangle \langle v, v \rangle - \langle u, v \rangle^2} ,$$

where $\mathcal{R}$ is the Riemann curvature tensor.

### H.3  COMPUTATION OF THE RIEMANN CURVATURE TENSOR

Our next task is to compare the Riemannian curvature of $\mathcal{M}$ with that of ambient space $\widetilde{\mathcal{M}}$. According to the definition of curvature tensor, we first give the relationship between the Riemannian connection $\nabla$ of $\mathcal{M}$ and $\widetilde{\nabla}$ of $\widetilde{\mathcal{M}}$:

$$\tilde{\nabla}_X Y = \nabla_X Y + \mathcal{B}(X, Y),$$

where the normal component is known as the second fundamental form $\mathcal{B}(X, Y)$ of $\mathcal{M}$. The second fundamental form uncovers the extrinsic structure of a manifold $\mathcal{M}$ relative to ambient space $\widetilde{\mathcal{M}}$. How the manifold is curved with respect to the ambient space is measured by the second fundamental form.

**Theorem 5** (The Gauss Equation (Lee, 2006)). *For any vector fields $X, Y, Z, W \in T\mathcal{M}$ the tangent bundle of $\mathcal{M}$, the following equation holds:*

$$\widetilde{\mathcal{R}}(X, Y, Z, M) = \mathcal{R}(X, Y, Z, W) - \langle B(X, W), B(Y, Z) \rangle + \langle B(X, Z), B(Y, W) \rangle$$

where $\widetilde{\mathcal{R}}$ is the Riemann curvature tensor of $\widetilde{\mathcal{M}}$ and $\mathcal{R}$ is that of $\mathcal{M}$. Riemannian curvature of the ambient space can be decomposed into two components. In this paper the ambient space is Euclidean space $\mathbb{R}^D$, so $\widetilde{\mathcal{R}}(X, Y, Z, W) = 0$. In this case, the Riemannian curvature of $\mathcal{M}$ is represented as:

$$\mathcal{R}(X, Y, Z, W) = \langle \mathcal{B}(X, W), \mathcal{B}(Y, Z) \rangle - \langle \mathcal{B}(X, Z), \mathcal{B}(Y, W) \rangle$$

To compute the value of the second fundamental form, we construct a local natural orthonormal coordinate frame $\left\{ \frac{\partial}{\partial x^1}, \cdots, \frac{\partial}{\partial x^d}, \frac{\partial}{\partial y^1}, \cdots, \frac{\partial}{\partial y^{D-d}} \right\}$ of the ambient space $\widetilde{\mathcal{M}}$ at point $p$, the restrictions of $\left\{ \frac{\partial}{\partial x^1}, \cdots, \frac{\partial}{\partial x^d} \right\}$ to $\mathcal{M}$ form a local orthonormal frame of $T_p\mathcal{M}$). The last $D - d$ orthonormal coordinates $\left\{ \frac{\partial}{\partial y^1}, \cdots, \frac{\partial}{\partial y^{D-d}} \right\}$ form a local orthonormal frame of $\mathcal{N}_p(\mathcal{M})$. Under the locally natural orthonormal coordinate frame, the embedding map $f$ is redefined as $f\left(x^1, x^2, \cdots, x^d\right) = \left[x^1, x^2, \cdots, x^d, f^1, \cdots, f^{D-d}\right]$, where $x \doteq \left[x^1, x^2, \cdots, x^d\right]$ are natural parameters. Then the second fundamental form $\mathcal{B}$ can be written as:

$$\mathcal{B}\left( \frac{\partial}{\partial x^i}, \frac{\partial}{\partial x^j} \right) = \sum_{\alpha=1}^{D-d} h_{ij}^{\alpha} \frac{\partial}{\partial y^{\alpha}}$$

with $h_{ij}^{\alpha}, (\alpha = 1, \cdots, D - d)$ being the second derivative $\frac{\partial^2}{\partial x^i \partial x^i}$ of embedding component function $f^{\alpha}$, which constitutes the Hessian matrix $H^{\alpha} = \left( \frac{\partial^2}{\partial x^2 \partial x^j} \right)$, correspondingly, the Riemann curvature tensor of $\mathcal{M}$ is represented as:

$$R_{iljk} = \sum_{\alpha=1}^{D-d} \left( h_{ik}^{\alpha} h_{lj}^{\alpha} - h_{ij}^{\alpha} h_{lk}^{\alpha} \right).$$

It follows that to compute the Riemann curvature of Riemannian submanifold $\mathcal{M}$, we only need to estimate the Hessian matrix of the embedding map $f$. The Hessian matrix estimation is described in Sec. E.

