# OpenReview forum: "Curved Data Representations in Deep Learning"
_ICLR.cc/2023/Conference — Submitted to ICLR 2023_

### Official Review · Reviewer_iQQx · 2022-10-24

**Confidence:** 4
**Correctness:** 4
**Technical Novelty And Significance:** 3
**Empirical Novelty And Significance:** 4
**Recommendation:** 8

**Clarity, Quality, Novelty And Reproducibility:**

This work is clear and carefully written. The results appear to me novel and of good-quality. I did not find any code submitted to check the reproducibility of the work. I noted the author's comment of releasing the code upon publication and encourage them to do so.


**Strength And Weaknesses:**

[+] Convincing empirical support for the characteristic trend of layer-wise curvature (Figure 1).

[+] Good evidence and discussion of the relationship between ID and curvature (Section 4.3), clarifying a potential misunderstanding in the literature.

[+] Reasonable choice of "relatively" efficient curvature measure (CAML, Li 2018)

[-] More discussion of how this work's results are consistent with prior theoretical studies (Cohen et al. 2020) is desired.

[-] Very limited conclusion given (single sentence). Please include a concluding section in the next revision.



**Summary Of The Paper:**

This work carries out an empirical study of the role of data manifold curvature in neural networks. The measure of curvature used is the principle curvature per sample, an algorithm for which is given by prior work. This measure is evaluated layer-wise throughout a network for several common CNNs architectures. It is observed that data evaluated with trained networks has a characteristic pattern: curvature increases, plateaus, then increases again. Furthermore it is found the gap between initial and final curvatures predicts generalization. Empirical evaluations of weight decay, regularization, and mixup training are also made.


**Summary Of The Review:**

Overall this work offers a thorough empirical investigation of the role of curvature in deep networks, similar in spirit to Ansuini et al (2019). This work offers a number of interesting insights and helps clarify some issues in the literature. Overall I recommend acceptance.

---

> ### Author Response · Authors · 2022-11-17
> **Response to Reviewer iQQx**
>
> We would like to thank Reviewer iQQx for positively commenting on our convincing empirical results, the choice of curvature estimate, and our discussion on the relationship between dimensionality and curvature. Below, we address the comments raised by Reviewer iQQx. Given the opportunity, we will be happy to incorporate the modifications listed below into a final revision.
>
> 1.  Relation to Cohen et al. 2020:
>
> In Cohen et al. (2020), the authors study the notion of an ensemble of manifolds, where each manifold represents same-class samples, e.g., in binary classification we have two manifolds. Then, the authors define the radius of the ensemble as a normalized value based on the total variance of its anchor points, i.e., points that characterize the separation between manifolds. While the radius by the definition of Cohen et al. (2020) is not a direct measurement of curvature, this measure is related to the geometry of the manifold. Then, the authors compute the radius along the layers of convolutional neural networks such as AlexNet and VGG. In general, they find a decrease in the radii and relate it to the classification capacity of CNN. Our work is specifically related to theirs in two aspects: In their Fig. 4, they find that the classification capacity increases significantly in the last layer of CNNs, which may be explained by our empirical findings of an increase in curvature. Similarly, in their Fig. 6b they present radius profiles for AlexNet and VGG for smooth 2-d manifolds (generated by image transforms) which are almost a mirror of the MAPC plots we present. Specifically, they show a sharp decrease in the first layer, followed by a plateau from most inner layers, and ending with a further decrease in the last few layers. We will add this discussion to the revision.
>
> 2. Limited conclusion
>
> Given the opportunity, we would be happy to extend the discussion in the paper to make the main message of the paper clearer. Our empirical setting is mostly related to that of Ansuini et al. 2019 and Doimo et al. 2020, and thus, we will discuss our findings with respect to these studies, although naturally, other works such as Poole et al. 2016 and Cohen et al. 2020 are also related. Therefore, we consider three data representations properties: intrinsic dimension (ID), the probability density of neighbors overlap with the output (PDN), and mean absolute principal curvature (MAPC). Intuitively, when the ID is high, then data points can move in more directions, and the opposite is true when the ID is low. If the PDN is low, then data points have not yet settled in their final destination, whereas when the PDN is high, then the majority of points are fixed in their position. Finally, low MAPC values mean that data points can move "cheaply" across the manifold, and if the MAPC is high, then some directions are expensive to move along. Aggregating the empirical findings and analysis from our paper and Ansuini et al. 2019 and Doimo et al. 2020, we arrive at the following understanding of deep convolutional neural networks: during the first layers of the network (corresponding to the increase in ID), the network allows data points to change their position on the manifold as fast as possible; this is achieved by the increase of ID providing more directions to move along, and low MAPC values which make travel across the manifold to be cheap. Then, during the intermediate layers of the network (corresponding to the plateau in MAPC), points still travel cheaply due to low MAPC values. However, during this phase the network gradually eliminates motion directions, yielding gradually better representations of latent information. Finally, during the last layers of the network (corresponding to the nucleation in Doimo et al. 2020, and the jump in MAPC in our paper), most points are fixed in their final destination; this is achieved by the network via learning a manifold with a very low ID, allowing data points to move in a limited amount of directions, as well as high MAPC which makes travel to be costly along high curvature directions. High PDN values in these layers reinforce the observation that most points reached their final destination. From a geometric viewpoint, manifolds throughout the intermediate layers of the network are quasi-Euclidean (i.e., close to flat), differing essentially in their ID, whereas the last manifolds are potentially "pointy" hyper-spheres, where clusters of same-class samples concentrate on "hills", and classification is obtained by hyper-planes separating different hills.

---

### Official Review · Reviewer_KXuv · 2022-10-25

**Confidence:** 4
**Correctness:** 3
**Technical Novelty And Significance:** 2
**Empirical Novelty And Significance:** 2
**Recommendation:** 5

**Clarity, Quality, Novelty And Reproducibility:**

Clarity and quality of the presentation are very good. So is reproducibility as the authors promise to release their codes.

- Section 3
  - Please include an explicit explanation as to how the work of Yu et al. (2018) relates to this discussion on density.
  - The intrinsic dimension can vary from one point to another in stratified manifolds; see, e.g., [Whitney conditions](https://en.wikipedia.org/wiki/Whitney_conditions).
  - I strongly recommend you tune down the claim in "yields a well-defined local patch of the manifold." The sampling approach is plausible, but it's unclear how it relates to the actual manifold.
  - Please define the "natural orthonormal coordinate frame"

- Section 4
  - $4.1: Please include an explicit explanation of how the theoretical studies of (Cohen et al., 2020), and empirical explorations on neural networks with random weights (Poole et al., 2016) are consistent with your empirical observations.
  - $4.6: the histograms confirm the increase in MAPC of the last layer, but does it explain it?

- Questions
  - What happens if you train using the augmented dataset, with a densely sampled neighborhood around each image? Could the initial increase in curvature be related to data density?
  - If I'm not mistaken, the paper only studies supervised learning. What would be the case for unsupervised learning?

**Strength And Weaknesses:**

- Strengths
  - Well-formulated and presented study of the curvature profile of intermediate latent representations
  - Empirical characterization of a common curvature profile for trained CNNs, with comparisons against untrained networks and using different regularizations
- Weaknesses
  - The work stays at a descriptive level. I appreciate the authors including the additional study in Appendix D, though more work is definitely needed.
    - The impact of the loss functions is not sufficiently studied. Perhaps the authors could consider a larger family of loss functions and regularizers, or include additional measures such as the ID, to help distinguish the behavior of different training strategies.

**Summary Of The Paper:**

The authors study the curvature profile of latent representations through the layers of deep convolutional neural networks, similar to prior works on the intrinsic dimensions pioneered by Ansuini et al. (2019). The authors propose a plausible approach to densely sample the neighborhood around each input image, as needed for the numerical computation of second derivatives.

**Summary Of The Review:**

The paper presents initiates the study of curvature profiles of latent representations through the layers of deep CNNs. Main findings uncover a common pattern of the curvature profiles for trained vs. untrained networks, that seems to persist using different regularizers. The paper is well-written which would help continue the research, especially with the release of the code.

---

> ### Author Response · Authors · 2022-11-10
> **Clarification**
>
> Thank you for your detailed review. Before we address all the remarks, a clarification would be much appreciated.
>
> Our work focuses on image classification using popular convolutional neural networks. For these architectures, classification is solved using the cross entropy loss function in the vast majority of cases, and thus our work uses that loss function. Could you elaborate what do you mean by a larger family of loss functions and regularizers and the inclusion of additional measures?

---

> > ### Comment · Reviewer_KXuv · 2022-11-12
> > **Follow up**
> >
> > Then perhaps the title of the paper could be more like: curvature of image representations + deep/neural/classification.
> >
> > My main reservation is the descriptive nature of the work without sufficient insights to help actually understand the observed phenomena. I wondered if changing more ingredients could help tease out more nuances. Please see the questions raised by reviewer revH under weaknesses; they had a better take than mine.
> >
> > One may consider a series of similar papers reporting on all sorts of measures of latent representation with similar/new patterns, e.g., curvature, topologies, concentration, symmetries, etc. What is the insight or message? How could this inform me when I train my next deep model and try to understand its limitations or make it better?
> >
> > It would be great if we can start by a better positioning of the reported empirical findings to the theoretical results of (Cohen et al., 2020), and possible the experiments by (Poole et al., 2016), as cited in $4.1

---

> ### Author Response · Authors · 2022-11-17
> **Response to Reviewer KXuv**
>
> We would like to thank Reviewer KXuv for finding our study to be well-formulated and for acknowledging our empirical characterization of curvature. Below, we address the comments raised by Reviewer KXuv. Given the opportunity, we will be happy to incorporate the modifications listed below into a final revision.
>
> 1. An explicit explanation of the relationship between Yu et al. (2018) to our work on density
>
> In Yu et al. (2018), the authors introduce the sampling scheme based on zeroing small singular values. In our work, we use the same procedure. We will modify the text to make this relation explicit.
>
> 2. Intrinsic dimension and stratified manifolds
>
> Our work assumes the underlying manifold is connected, and thus there is a single intrinsic dimension value. A similar assumption is made by Ansuini et al. (2019) and Birdal et al. (2020).
>
> 3. Tone down claim "yields a well-defined patch of the manifold"
>
> Thank you. We will re-phrase that sentence.
>
> 4. Define "natural orthonormal coordinate frame":
>
> The local natural orthonormal coordinate frame is defined as the basis associated with the tangent space and normal space at a point $p$ of the manifold. In practice, the coordinate frame is generated using PCA, where the first $d$ coordinates (associated with the most significant modes, i.e., largest singular values) represent the tangent space, and the rest encode the normal space. We will add this definition to the text.
>
> 5. Explicit explanation of how Cohen et al. (2020) and Poole et el. (2016) are consistent with our work
>
> In Poole et al. (2016), the authors study how the curvature of the manifold and the decision boundary change along the layers of the network. For data representing a circle, they find that the curvature of the manifold increases with depth, which is consistent with our results. In Cohen et al. (2020), the authors study the notion of an ensemble of manifolds, where each manifold represents same-class samples, e.g., in binary classification we have two manifolds. Then, the authors define the radius of the ensemble as a normalized value based on the total variance of its anchor points, i.e., points that characterize the separation between manifolds. While the radius by the definition of Cohen et al. (2020) is not a direct measurement of curvature, this measure is related to the geometry of the manifold.
>     Then, the authors compute the radius along the layers of convolutional neural networks such as AlexNet and VGG. In general, they find a decrease in the radii and relate it to the classification capacity of CNN. Our work is specifically related to theirs in two aspects: In their Fig. 4, they find that the classification capacity increases significantly in the last layer of CNNs, which may be explained by our empirical findings of an increase in curvature. Similarly, in their Fig. 6b they present radius profiles for AlexNet and VGG for smooth 2-d manifolds (generated by image transforms) which are almost a mirror of the MAPC plots we present. Specifically, they show a sharp decrease in the first layer, followed by a plateau from most inner layers, and ending with a further decrease in the last few layers. We will add this discussion to the revision.
>
> 6. Do histograms explain the increase in MAPC?
>
> Thank you. Indeed, the histograms confirm the results shown with the MAPC measure. We will modify the text and replace the word "explain" with "confirm".
>
> 7. What would be the case for unsupervised learning?
>
> Thank you for proposing this idea. Indeed, our work focuses on the supervised setting. In the related work, we mention the work of Shao et al. (2018) which studied the curvature of manifolds in variational autoencoder networks and showed the curvature of image data is almost flat. We leave further explorations of curvature analysis of unsupervised tasks for future work.

---

### Official Review · Reviewer_8GhN · 2022-10-25

**Confidence:** 4
**Correctness:** 3
**Technical Novelty And Significance:** 3
**Empirical Novelty And Significance:** 3
**Recommendation:** 5

**Clarity, Quality, Novelty And Reproducibility:**

Clarity: Overall it is quite good but I find the part on curvature estimation (in Sec. 3) to be quite confusing.

Quality: I am not entirely convinced that the paper is measuring the right curvature. Also the conclusion that curvature increases from input to output seems counter-intuitive.

Novelty: An analysis of curvature profile in deep networks is novel to the best of my knowledge.

**Strength And Weaknesses:**

# Strength

Manifold modeling provides an interesting perspective towards deciphering the role of multiple layers in a deep neural network. Previous work of Ansuini et al. NeurIPS'19 provides such an analysis using the intrinsic dimension of the manifold. This submission provides a similar study to that of Ansuini but with the manifold curvature instead of dimension. Hence, the result provides a more comprehensive understanding of the role of layers from a low-dimensional modeling perspective.

The paper is generally well-written though I find it hard to understand the technical part of it (on estimating curvature).

# Weakness

**Validity of the method for obtaining data density.** Part of the difficulty in evaluating curvature compared to dimension is that the former may require a much denser sampling of the manifold (intuitively). The paper addresses this challenge by a special kind of data augmentation described in Sec. 3, where (roughly) each image is performed a PCA and selected trailing components are dropped.

I am not sure if this is a valid way of increasing data density and some more explanation or study seems needed. Specifically, for manifold of natural images, curvature may be caused by many things such as translation, rotation, illumination, etc, but by the proposed method for data augmentation, only curvature caused by this specific way of augmentation is studied. In fact, the effect of such a data augmentation is a bit unclear in terms of what it does to the data manifold. It may be interpreted as adding some small noise to some selected directions in the ambient image space, but it appears that this will actually generate images that are outside of the image manifold, hence it is unclear what curvature means for them.

Question: What if one simply calculates curvature using only the original images without using any other augmentation for increasing density?

**Clarity.** I find the explanation on *Curvature estimation* in Section 3 to be quite confusing, that I was not able to follow it. E.g.,

- the embedding map f: what is it? it is used in eq. (1) but seems not explained at all.
- x_i: this is used in both eq. (1) and 7 lines below eq. (1). Are they the same?
- K: is this taken to be 1024?
- partial / partial x^1 (6 lines below Eq. (1)): Is this the partial gradient? of what? also what is x^1?

**So, deep network is not flattening the manifold?** One of the main conclusions is that the curvature increases from the first to the last layer of a deep neural network. This appears to be very surprising and what one may have expected is that the deep network flattens the manifold gradually so that the curvature should be a decreasing function. Do I miss anything here or is there an explanation for this?

**Negative curvature.** I am not entirely understanding results in Fig. 5 in that the curvature here can be negative while in all previous plots, curvature is always positive. How should I understand why negative curvature may appear here?

# Other comments / suggestions / questions

**Toy data.** It may be interesting to perform the analysis using some toy dataset with nonlinear manifold distribution. Then, one may examine how e.g. an MLP flattens the manifold and whether the observation aligns with what is observed with natural images.

**untrained network exhibit a different curvature profile**: here the experiment is only conducted on VGG. I am wondering what would happen for ResNets? In particular, because ResNet has a shortcut, if the residual branch is initailized to be small enough, then all layers should have very similar features hence curvature. Is this what is observed in practice?

**Some missing references.**

- Manifold-aware learning: Work on designing objectives and architectures for learning low-dimensional structures, e.g.

Chan, Kwan Ho Ryan, et al. "Redunet: A white-box deep network from the principle of maximizing rate reduction." J Mach Learn Res 23.114 (2022): 1-103.

- Manifold-aware analysis:

Buchanan, Sam, Dar Gilboa, and John Wright. "Deep networks and the multiple manifold problem." arXiv preprint arXiv:2008.11245 (2020).

**Small issues**

- Sec. 4.1: MAPC is coming out of nowhere as it is never explained before.
- Sec. 4.1, "we observe that MAPC is ...": not clear what I should be looking at.
- Sec. 4.1, "during the transition between first and second layers": does this mean from the output of the first layer to the output of the second layer?





**Summary Of The Paper:**

This paper is about understanding the functionality of layers in a deep neural network by examining the curvature of the low-dimensional manifold associated with the feature maps at different layers. To evaluate the curvature, one has to have sufficiently dense samples on the manifold and this is achieved by adding small perturbations to the natural images. The paper makes the following observations. 1) curvature along layers increases drastically from the first to the second layers, and from the second last to the last layer, while stays stable for all other layers, 2) curvature gap on final layer predicts generalization performance, 3) curvature and intrinsic dimension don't necessarily correlates, 4) the effect of regularization on curvature profile.

**Summary Of The Review:**

The paper provides an interesting study of deep neural networks from the perspective of manifold curvature, but I am not entirely convinced that their method is measuring the right curvature by their data sampling via PCA. I also find one of the main conclusions to be counter-intuitive and there is a lack of discussion. Finally, there is a clarity issue with the discussion of curvature estimation.

---

> ### Author Response · Authors · 2022-11-17
> **Response to Reviewer 8GhN**
>
> We would like to thank Reviewer 8GhN for identifying the importance of our results and for finding our paper to be well-written. Below, we address the comments raised by Reviewer 8GhN. Given the opportunity, we will be happy to incorporate the modifications listed below into a final revision.
>
>
> 1. Is SVD a valid way of increasing data density?
>
> Indeed, curvature estimates for high-dimensional and sparse point clouds are extremely noisy and unreliable. To alleviate this issue, we aimed to (locally) increase the density of the data manifold. Our choice to use SVD is well-motivated from a differential geometry viewpoint. Specifically, the SVD procedure we described in Sec. 3 is closely-related to computing a first-order approximation of the manifold at a point, and sampling at the neighborhood of the point (see e.g., Sec. 2.4 in Curvature-aware Manifold Learning by Li 2018). We agree with Reviewer 8GhN that the sampled points may slightly deviate from the data manifold. The deviation can be bounded by the absolute value of the modified singular values (which are close to zero in practice).
>
> In addition to the theoretical justification we provide for the SVD procedure, we also experimented with up-sampling approaches based on standard image transforms, e.g., translation, shear, and rotation. That is, we train the networks as before, but during inference, we feed a point with its neighborhood based on image transforms. We use the following image transformations: rotation=10, shear=(10,10,10,10), and translation=(0.1,0.1). Remarkably, we find for CIFAR10 on ResNet and VGG architectures a characteristic profile akin to the curvature profiles we report e.g., in Fig. 1, with one qualitative different feature. The curvature profiles associated with image transforms exhibit a significant drop in the penultimate layer, whereas curvature profiles as reported in Fig. 1 do not present this characteristic. We believe it is related to the low-dimensionality of the data (as governed by the image transforms) allowing to form a low curvature manifold in comparison to our SVD-based sampling which yields closer points, but with potentially more intrinsic dimensions, making it harder to encode using very low MAPC values. We will add this discussion and results to the revision.
>
> 2. What if curvature is calculated directly on non-augmented data?
>
> As discussed in Sec. 3 in the paragraph about data density, one needs dense neighborhoods when estimating high-order derivatives on high-dimensional data. If the data represents a sparse sampling of the manifold, then curvature estimates are extremely noisy and unreliable. Our proposal to augment neighborhoods of images with artificially generated samples is meant to increase the density per sample, and as a result, to improve the robustness of our curvature estimates.
>
> 3. Confusing explanation in Sec. 3
>
> We will improve the text in Sec. 3. The embedding map $f$ is the transformation that maps the low-dimensional samples to their high-dimensional counterparts (the latter are the observed images). In a sense, $f$ maps the "essential" information in the image to a pixel-wise representation that holds redundant information. Added to the main text.
>
> Yes, $x_i$ in Eq. (1) and 7 lines below are the same $x_i$.
>
> Yes. $K$ is a hyper-parameter which in our experiments is taken to be $1024$.
>
> In differential geometry, it is common to represent the axes of a coordinate frame with the notation $\frac{\partial}{\partial x^i}$, where $x^i$ in this context is the $i$-th coordinate.
>
> 4. Deep networks do not flatten the manifold?
>
> Indeed, one of the main conclusions of our work is that the curvature of the manifold increases with depth. We emphasize that in several existing works in the literature, it is claimed that neural networks flatten with depth. However, these works (e.g., Poole et al. 2016) make these claims with respect to the curvature of the *decision boundary* and not the curvature of the manifold (which we study in our work). The curvature of the decision boundary and that of the manifold are two very different notions of curvature. Our work does not contradict previous findings showing that the decision boundary becomes flat with depth, but it rather complements previous work with empirical results related to the curvature of the data manifold.
>
> 5. Negative curvature
>
> In general, the principal curvatures (i.e., eigenvalues of Hessian) need not be positive. Fig. 5 shows the distribution of principal curvatures as obtained by computing the eigenvalues of the various Hessian matrices. In Figs. 1-4, we show the mean *absolute* principal curvature (MAPC), and thus these plots do not show negative values.

---

> > ### Comment · Reviewer_8GhN · 2022-11-17
> > **Followup questions**
> >
> > Thanks for the response.
> >
> > - I am still not entirely convinced by data augmentation via SVD. When you say "sampling at the neighborhood of the point", does that mean a neighborhood *on the manifold*, or simply a neighborhood measured by Euclidean distance? "the sampled points may slightly deviate from the data manifold": it is okay to deviate as long as the major variance is within the manifold tangent plane. Can the authors help to confirm whether augmentation via SVD produces images on the manifold of natural images?
> >
> > - Deep networks do not flatten the manifold & toy data.
> >
> > My personal opinion is that toy data provides much insight into deep neural networks, even if the behavior does not necessarily align with those on real data. This is particularly the case for this paper since the main message of the paper that deep networks do not flatten the manifold is quite surprising, and I have doubts on whether SVD causes this. Using toy data does not have the issue of sampling density hence does not need to perform SVD, which may help to provide more insights.

---

> > > ### Author Response · Authors · 2022-11-27
> > > **Followup response**
> > >
> > > While the structure of the manifold is unknown, the SVD method produces samples that are extremely close in the Euclidean distance sense justifying our assumption that the new samples may slightly deviate from the data manifold. In our revised version we included a visual comparison between the samples generated using the SVD method (see Fig. 9). It is notable that the generated images are almost identical. Thus they probably lay on the manifold of natural images.
> > >
> > > During the rebuttal, we added CAML computations where instead of SVD we employ standard image transformations. The plots obtained with image transforms are similar qualitatively and quantitatively to the characteristic profiles resulting from SVD-based sampling (please see the revised version, Fig. 8 ). These results reinforce the main results in the paper that the CAML profile is characteristic to deep convolutional neural networks as observed using manifold neighborhoods based on SVD and image transform estimations.

---

> > > > ### Comment · Reviewer_8GhN · 2022-11-29
> > > > **Sampling on manifold**
> > > >
> > > > > "the SVD method produces samples that are extremely close in the Euclidean distance sense", "It is notable that the generated images are almost identical. "
> > > >
> > > > I don't think these are valid justification that the paper is measuring the right manifold. Here is the picture I have in mind: Suppose we have a low-dimensional manifold, say a curve in 2D. Then we can generate a collection of data points by adding random Gaussian noise to a point on the manifold. Then, if the variance of the noise is small enough, this collection of points satisfy what the response above described, i.e., they can be extremely close to the manifold in the Euclidean distance sense, and they are almost identical. However, the curvature computed from this collection of data points has nothing to do with the curvature of the manifold at all.
> > > >
> > > > > "we added CAML computations where instead of SVD we employ standard image transformations"
> > > >
> > > > Thanks, this is very helpful. But it is restricted to simple affine transformations, and that the characteristic profile is not entirely the same as what was observed with SVD augmentation. This already suggests that what "curvature" you are measuring (i.e., curvature from SVD or affine transform) matters for drawing the conclusion. My suggestion is that given its title, the paper should perhaps be measuring curvature from varying kinds of variations, say affine transform and other commonly used data augmentations for network training, and understand the commonality and differences of the curvature profile from them.

---

> > > > > ### Author Response · Authors · 2022-12-09
> > > > > **Response to sampling on manifold**
> > > > >
> > > > > ``However, the curvature computed from this collection of data points has nothing to do with the curvature of the manifold at all.''
> > > > >
> > > > > This is incorrect. Specifically, the well-known Steiner formulas describe variations in the area and volume of surfaces in terms of the curvature of the surface (see e.g., Geometric Measure Theory by Federer, 1996).
> > > > >
> > > > > ``But it is restricted to simple affine transformations...''
> > > > >
> > > > > Affine transformations are one of the most common data augmentations applied to images. In particular, affine transformations are studied thoroughly in works similar to ours as they are commonly assumed to generate images on the data manifold of natural images. Indeed, we agree that there are small differences between the MAPC plots obtained with affine transformations in comparison to those obtained with SVD. However, these variations do not change the results qualitatively. We are happy to change the title to better describe the contents of our work. For instance, perhaps a more accurate title would be: Curvature Analysis of Image Data Representations via Approximate Local Patches of Manifolds.

---

> ### Author Response · Authors · 2022-11-17
> **Response to Reviewer 8GhN (2)**
>
> 6. Toy data
>
> Thank you for proposing this idea. In our experience, observations and insights obtained on toy data with toy neural networks typically do not extend to real-world datasets and state-of-the-art architectures. Therefore, it is not clear what can be gained from an experiment involving toy components. Our work focuses specifically on challenging datasets and common architectures which are used in practice, similar to studies on the intrinsic dimension, e.g., Ansuini et al. (2019).
>
> 7. Untrained networks
>
> While ResNet models have shortcut connections, their nonlinear nature due to activation functions (applied also on shortcuts) modifies the geometric properties of data manifolds, even for untrained networks. In particular, we run this experiment again on ResNet-50 and ResNet-101 and obtained qualitatively similar profiles. The MAPC values we got for ResNet-50 and for ResNet-101 are small at the first layer ($\approx$1e-6), increase and remain relatively constant in the intermediate layers ($\approx$1e-1), and then decrease abruptly in the last layer ($\approx$1e-10). These results confirm that untrained ResNets behave similarly to VGG networks with respect to the MAPC measure, where the main difference is at what layer the drop in MACP occurs. In ResNet the decrease occurs in the last layer, whereas in VGG networks it varies, but generally occurs around the middle of the network. We will add the graphs to the revised version.
>
> 8. Missing references
>
> We will add the missing references in the revised version.
>
> 9. Small issues
>
> We will fix all small issues in the revised version.

---

### Official Review · Reviewer_revH · 2022-11-04

**Confidence:** 4
**Correctness:** 3
**Technical Novelty And Significance:** 2
**Empirical Novelty And Significance:** 2
**Recommendation:** 3

**Clarity, Quality, Novelty And Reproducibility:**

The paper is clearly written. The authors have made the experimental approach explicit and easy to understand. The authors explain all the figures in good detail. On the other hand, the writing focuses too much on what is plotted in the figures and misses the connection between the experiments and how they contribute to the larger picture. As a result, it reads more like a few separate experiments without having a coherent message.

**Strength And Weaknesses:**

Strength

1. The question this paper study is an interesting and important one. The geometric properties of the data will affect the trainability of the network. Studying how these geometric properties change across layers can help us understand what types of features are learned or preferred. Previous literature focuses more on the intrinsic dimensionality of the data manifold, and this paper investigates how the curvature behaves for convolution neural networks, which definitely worth studying.

2. The paper proposes a single measure MAPC that captures the overall curvature of the data manifold, and provides empirical observations of a three-stage change of the MAPC across layers: an initial increase, followed by a stable phase in the middle, and another final increase. The observation is stable across different network and training setups and datasets so are quite convincing. The authors also relate the normalized MAPC gap to the model accuracy which can have further implications.


Weakness:

Although the paper provides many nice experiments and observations, the authors do not put them into a coherent picture and provide further justification for the observed phenomena, making it hard to grasp the contribution of the paper and how one should build understanding upon all the figures.

1. Missing justification of the MAPC. As the authors have pointed out, there is a difference between the curvature of the decision boundary and the curvature of the data manifold. MAPC is a metric of the latter and the authors failed to provide a justification over why the curvature of the whole manifold is preferred and why MAPC is a good measurement for it. The authors do discuss the choice of curvature metrics in appendix A, saying all metrics exhibit similar behaviors and claim using MAPC "due to the lack of a canonical metric". and MAPC is a metric for the latter. This arbitrary choice of curvature measures fails to convince readers why MAPC is able to capture key geometric information of the data.

2. The main message of the paper is unclear. There are many experimental observations but the authors fail to put them into a complete picture. Two important findings in the paper are that there is a three-stage change of the curvature along layers and that the normalized MAPC gap is correlated with the model accuracy. However, the authors use most space describing the figures without providing enough explanations of where the phenomena come from. The three-stage phenomenon is actually counterintuitive. It is more natural to consider the network "flattening" features through layers instead of making them curvier. One explanation that one can come up with is that the network flattens data along decision boundaries, and compress data elsewhere, resulting in the increase of MAPC due to the compression and averaging effect. The arguments in the paper would be much stronger if the observations can be supported by detailed reasoning and further empirical justifications.

3. The missing explanation also makes it hard to utilize the findings in the paper. For example, the authors argue that the normalized MAPC gap is correlated with model accuracy, but does that mean we would like to learn networks that explicitly encourage a large curvature gap between the last two layers? This unclarity of future direction also compromises the importance of the findings.

Other than the contribution of the paper, I also have questions about the computation of MAPC

4. The data lies in a high-dimension space, making the estimation of the data manifold hard. In the paper, the authors propose a data augmentation method to get enough local data by blurring (removing information in some of the small singular value spaces). This process is done for each image separately and thus it may not preserve the manifold information. The curvature information based on this data augmentation regime is more likely to capture how robust the network is towards the small singular value spaces.

5. To compute the curvature or MAPC, one needs to fix a dimensionality. In the paper, the dimensionality is computed through the TwoNN algorithm for each layer. On the other hand, one can imagine that the curvature information is highly correlated with dimensionality. By allowing a larger dimensionality, one can fit the data with a much smaller curvature. The discussion about this relationship between dimensionality and curvature is missing in the paper.
In the discussion session, the authors mention that "as computation proceeds, samples concentrate near their same-class samples in highly-curved peaks, facilitating separation between clusters". Will this indicates that even though the data is highly curved, as it is concentrated, if one views it in a higher dimensional space, the curvature can actually be small?


**Summary Of The Paper:**

The paper studies how the curvature of the data manifold changes across different layers of the trained networks. It proposes the mean absolute principle curvature (MAPC) that characterizes the averaged curvature of the data representations, and shows that there is a three-stage behavior of the MAPC along layers: an initial increase, a long phase of a plateau, and another final increase. This observation is consistent among different network and training setups and is not observed for untrained networks. The authors further show that the curvature gap between the last two layers is correlated with the performance of the network, and can be used to predict the generalization ability of the networks.

**Summary Of The Review:**

The paper studies a fairly interesting problem that is worth further efforts to investigate. The authors provide interesting experiments showing a three-stage behavior of the measurements of the MAPC and the relationship between the normalized MAPC gap and the performance of the network. However, the authors fail to provide enough explanation of either the importance of the MAPC or the reasoning for the three-stage behavior. The reviewer considers the paper to be below the quality of acceptance.

---

> ### Author Response · Authors · 2022-11-17
> **Response to Reviewer revH**
>
> We would like to sincerely thank Reviewer reVH for their detailed and thorough review. Many of the points raised by Reviewer reVH are related to choices we made in our research and its exposition. We hope that in our response, we address the comments in their entirety. Our response and the modifications and changes we propose below can be incorporated into a final revision within the time frame of ICLR review process, given the opportunity.
>
> 1. missing justification of the MAPC.
>
> We will be happy to provide a more detailed justification for studying the curvature of data manifolds via MAPC, in addition to what is currently detailed in the manuscript. Our introduction and related work sections motivate and justify our focus on studying the curvature of data manifolds. To the best of our knowledge, most existing work considers curvature in the context of the decision boundary, which is a different object from the data manifold. The few works which analyze the data manifold in terms of curvature use unreliable geodesic estimates (Brahma et al. 2015) and focus on the comparison of two networks (Yu et al. 2018).  In fact, there is no extensive and systematic study that characterizes the curvature profile along layers of deep CNN, similar to our manuscript. Moreover, the recent papers of Ansuini et al. (2019) and Doimo et al. (2020) provide an extensive empirical investigation in terms of the intrinsic dimension and density evolution, respectively. These studies emphasize the need to have a more complete picture of the properties data representations exhibit. Thus, toward bridging these gaps in the literature, we investigate the curvature of data manifolds arising in deep convolutional neural networks.
>
> Why MAPC? principal curvatures uniquely define the curvature of a manifold at a point. From a Riemannian geometry viewpoint, our choice to focus on principal curvatures is well-justified, and their computation is backed by existing work (Li 2018). For a $d$-dimensional manifold, we have $d$ principal curvatures per point of the manifold. Therefore, we stand by our claim that there is no canonical curvature (scalar) metric, as we generally have in practice $N \cdot d$ values to consider, where $N$ is the number of data points. The mean absolute principal curvature (MAPC) can be viewed as a closely related variant of the mean curvature of *hypersurfaces*, which is a local extrinsic measure for curvature (see e.g., A comprehensive introduction to differential geometry by Spivak, Vol. 4). Our "ablation" study in Appendix A and Fig. 6 further shows that other principal curvature variants yield qualitatively similar observations as the MAPC. Finally, our overarching objective is to study the curvature properties of data representations. In this context, we propose MAPC as a reasonable estimate to consider and conduct our analysis based on it. The MAPC measure gains its motivation and justification from the empirical results we provide and their consistency across different networks and datasets. From this perspective, any other measure (e.g., intrinsic dimension or density evolution) is as arbitrary as the MAPC. However, we believe that if a certain measurement reveals new findings and/or backs previous findings, it warrants its study. Our paper shows that the MAPC is such a measure as its values for data manifolds correspond to existing work in the literature, and additionally, it brings new observations to our attention. Importantly, we do not claim that MAPC is the single curvature measurement one should consider. Indeed, we also show the principal curvature histograms (Fig. 5) as well as other average curvature measurements (Fig. 6). Our claim is somewhat different: we advocate the study of MAPC as it is highly consistent and leads to interesting insights on challenging image benchmarks using state-of-the-art deep convolutional neural networks.

---

> ### Author Response · Authors · 2022-11-17
> **Response to Reviewer revH (2)**
>
> 2. unclear main message.
>
> Given the opportunity, we would be happy to extend the discussion in the paper to make the main message of the paper clearer. Our empirical setting is mostly related to that of Ansuini et al. 2019 and Doimo et al. 2020, and thus, we will discuss our findings with respect to these studies, although naturally, other works such as Poole et al. 2016 and Cohen et al. 2020 are also related. Therefore, we consider three data representations properties: intrinsic dimension (ID), probability density of neighbors overlap with the output (PDN), and mean absolute principal curvature (MAPC). Intuitively, when the ID is high, then data points can move in more directions, and the opposite is true when the ID is low. If the PDN is low, then data points have not yet settled in their final destination, whereas when the PDN is high, then the majority of point are fixed in their position. Finally, low MAPC values mean that data points can move "cheaply" across the manifold, and if the MAPC is high, then some directions are expensive to move along. Aggregating the empirical findings and analysis from our paper and Ansuini et al. 2019 and Doimo et al. 2020, we arrive at the following understanding of deep convolutional neural networks: during the first layers of the network (corresponding to the increase in ID), the network allows data points to change their position on the manifold as fast as possible; this is achieved by the increase of ID providing more directions to move along, and low MAPC values which make travel across the manifold to be cheap. Then, during the intermediate layers of the network (corresponding to the plateau in MAPC), points still travel cheaply due to low MAPC values. However, during this phase the network gradually eliminates motion directions, yielding gradually better representations of latent information. Finally, during the last layers of the network (corresponding to the nucleation in Doimo et al. 2020, and the jump in MAPC in our paper), most points are fixed in their final destination; this is achieved by the network via learning a manifold with a very low ID, allowing data points to move in a limited amount of directions, as well as high MAPC which makes travel to be costly along high curvature directions. High PDN values in these layers reinforce the observation that most points reached their final destination. From a geometric viewpoint, manifolds throughout the intermediate layers of the network are quasi-Euclidean (i.e., close to flat), differing essentially in their ID, whereas the last manifolds are potentially "pointy" hyper-spheres, where clusters of same-class samples concentrate on "hills", and classification is obtained by hyper-planes separating different hills.
>
> 3. utilize findings.
>
> Our work complements previous studies on manifold properties of data representations arising in deep convolutional neural networks (specifically, Ansuini et al. 2019 and Doimo et al. 2020). With the results reported in our paper, researchers and practitioners receive a more complete picture regarding the characteristics of data representations throughout layers of the network. How should one utilize our findings? this is an important and deep question that requires a separate study, and thus it is beyond the scope of the current paper. Should one encourage a large curvature gap between the last two layers? unfortunately, we do not know yet. The high correlation between such a feature and a small generalization gap as shown in our findings definitely motivates such a future study. However, it is almost certainly not a sufficient condition, i.e., say one would promote training toward a large curvature gap (but no other penalties), then we believe it is highly unlikely that the trained network outperforms SOTA approaches. Nevertheless, investigating a new training procedure that incorporates findings from *all* empirical and theoretical studies (e.g., Ansuini et al. 2019, Cohen et al. 2020, and others) including ours into a new training procedure, is an exciting research direction we are actively working on.

---

> > ### Comment · Reviewer_revH · 2022-12-03
> > **Missed reasoning from empirical observations to the claim of the paper**
> >
> > I thank the authors for their detailed response.
> >
> > I consider my main concern is that the paper failed to provide enough evidence and reasoning to support its claim. From what I understand from the paper and from the response, the authors are trying to provide a detailed story of how neural network training is happening and how the feature changes along layers. On the other hand, what is shown in the experiment is an observation of how MAPC and ID change. A nontrivial reasoning about how the phenomena of MAPC and ID can be related to the story of network training that the authors are claiming here is missing.
> >
> > To be more specific, what is observed in the experiments is the three stages of MAPC change. Even if we assume this measurement is accurate, how does such an observation show "data points to change their position on the manifold as fast as possible", "points still travel cheaply due to low MAPC values", and "most points are fixed in their final destination". MAPC is a single measurement proposed in the paper, so it would require further experiments or reasoning to demonstrate that MAPC can be related to properties like "points traveling". In my original review, this missed reasoning is what causes the confusion and makes me unclear about the justification of MAPC and the main message of the paper. I don't think this is well addressed in the response.

---

> > > ### Author Response · Authors · 2022-12-09
> > > **Response to Reviewer revH**
> > >
> > > Thank you for continuing the discussion.
> > >
> > > Unfortunately, there still seems to be a misunderstanding of our results and their significance. Our paper does not analyze the training of convolutional neural networks. The only result related to training is shown in Fig. 11, where we show the correlation between MAPC gap and network performance as the network trains. All the other results in the paper are related to *trained* networks, and we analyze the trajectories of inputs as they are being processed by the network layers. Thus, when we describe the traveling of points, we mean the distribution of data points on the manifold when moving from one layer to its subsequent layer. The properties of the layer's transformation and the properties of the data manifold determine how points move along layers. In this context, low MAPC values indicate that the transformation applied by the layer can change the position of the points on the manifold relatively equally in all directions.

---

> ### Author Response · Authors · 2022-11-17
> **Response to Reviewer revH (3)**
>
> 4. why data augmentation with SVD?
>
> Indeed, curvature estimates for high-dimensional and sparse point clouds are extremely noisy and unreliable. To alleviate this issue, we aimed to (locally) increase the density of the data manifold. Our choice to use SVD is well-motivated from a differential geometry viewpoint. Specifically, the SVD procedure we described in Sec. 3 is closely-related to computing a first order approximation of the manifold at a point, and sampling at the neighborhood of the point (see e.g., Sec. 2.4 in Curvature-aware Manifold Learning by Li 2018). We agree with Reviewer 8GhN that the sampled points may slightly deviate from the data manifold. The deviation can be bounded by the absolute value of the modified singular values (which are close to zero in practice).
>
> In addition to the theoretical justification we provide for the SVD procedure, we also experimented with up-sampling approaches based on standard image transforms, e.g., translation, shear, and rotation. That is, we train the networks as before, but during inference, we feed a point with its neighborhood based on image transforms. We use the following image transformations: rotation=10, shear=(10,10,10,10), and translation=(0.1,0.1). Remarkably, we find for CIFAR10 on ResNet and VGG architectures a characteristic profile akin to the curvature profiles we report e.g., in Fig. 1, with one qualitative different feature. The curvature profiles associated with image transforms exhibit a significant drop in the penultimate layer, whereas curvature profiles as reported in Fig. 1 do not present this characteristic. We believe it is related to the low-dimensionality of the data (as governed by the image transforms) allowing to form a low curvature manifold in comparison to our SVD-based sampling which yields closer points, but with potentially more intrinsic dimensions, making it harder to encode using very low MAPC values. We will add this discussion and results to the revision.
>
> 5. fixed dimensionality
>
> There seems to be a potential misunderstanding about the relation between dimensionality and curvature in the context of our study. We investigate data manifolds of *trained* deep CNN. Thus, we assume that the underlying manifolds are fixed as training is finished. Under this assumption, it is not clear from a theoretical viewpoint what you mean by "allowing a larger dimensionality". When training ends, the intrinsic dimension, curvature, and other properties of data manifolds are *fixed*. In practice, we need to compute the principal curvatures of the manifold, which requires the use of an estimate of the intrinsic dimension. We propose to use TwoNN because it is one of the best algorithms to estimate the ID of high-dimensional data, and in addition, it makes our empirical results consistent with the work of Ansuini et al. (2019) and others (e.g., Zhu et al. 2018) which use TwoNN as well.

---

> > ### Comment · Reviewer_revH · 2022-12-03
> > **Questions for the SVD**
> >
> > Question about the data augmentation:
> >
> > The SVD approach is still questionable here. Please see Reviewer 8GhN's comments and address them properly. My main question is that the data augmentation approach is done for each data point separately. How can one get the curvature of the image manifold by looking at a single point on the manifold? In the authors' response to Reviewer 8GhN's comments, they mentioned Figure 9 in the paper. Actually, in Figure 9, non of the newly generated image is visually different from the original image, so they can be viewed as the original image with indiscernible noise. How can one achieve accurate curvature estimation of the image manifold based on such localized data augmentation?
> >
> > About dimensionality:
> >
> > My point for this comment is that even for a fixed noisy manifold, there may not be a canonical dimension. For example, consider a space-filling curve like Peano curves. They are one-dimensional with crazily high curvature, but if one views them as a two-dimensional manifold, the curvature would be zero. So even for a fixed trained manifold, the curvature could be wildly different by considering different dimensions.

---

> > > ### Author Response · Authors · 2022-12-09
> > > **Response on SVD and dimensionality**
> > >
> > > SVD:
> > > We would like to point out that curvature is a local property of a point on the manifold. In practice, one estimates curvature using an approximate patch of the manifold. To approximate the patch, we used many points, generated by the SVD approach and using affine transformations. Thus, your comment regarding us using a single point to "get the curvature" is incorrect. Moreover, the patch is generated at the input level, and then it is propagated through the nonlinear layers of the network. Therefore, having a local patch at the input level, does not mean it maintains the same level of locality at later layers. In practice, the points shift their position on the manifold and typically stretch away from the center point. We are happy that you agree that the patch we generate using SVD is local as is shown in Fig. 9. This is exactly the message we wanted to deliver. To generate accurate curvature estimates, it is preferable to obtain localized patches. The distance comparison in Fig.8B shows that the patches we compute with SVD are more localized than the patches with affine transformations, and thus are expected to yield better curvature estimates.
> > >
> > > Dimensionality:
> > > We believe that you are concerned about the sensitivity of curvature computations, conditioned on the actual intrinsic dimension value provided to the CAML procedure. As we discussed in the paper and during the rebuttal, our work strives to align with prior work in the field. In particular, we follow Ansuini et al. in terms of the evaluation setup, and thus, we use the same intrinsic dimension algorithm as they used. This makes the results in our paper and the analysis consistent with prior work in the field. It might be that other approaches to computing the intrinsic dimension will yield different results, which may potentially change some of the results in our paper. However, this may be also true for the Ansuini et al. paper.

---

### Author Response · Authors · 2022-11-27
**Future discussion**

We would like to thank again the reviewers for their detailed and helpful reviews. Additionally, we encourage the reviewers to raise any remaining issues, and to ask questions regarding our work. We would be happy to clarify points that are still unclear.

---

### Decision · Program_Chairs · 2023-01-20

**Decision:**

Reject

**Justification For Why Not Higher Score:**

As described above, the paper presents interesting findings on curvature of feature representations. In current form, there remain concerns both with the paper's approach (what is the object whose curvature is being measured) and the lack of a preliminary explanation of the origin of the curvature phenomenon and how it helps classification.

**Justification For Why Not Lower Score:**

N/A

**Metareview: Summary, Strengths And Weaknesses:**

The paper studies curvature of feature representations in deep neural networks. It observes two phenomena: first, estimated curvatures exhibit a characteristic profile, in which curvature increases at the first layer, is relatively stable at intermediate layers, and then increases again at the final layer. Second, a certain normalized curvature gap, which measures the curvature increase at the final layer, is positively correlated with network performance on test data. The characteristic curvature profile seems to be a product of network training, as it is not observed in untrained networks. The paper also presents a range of experiments, showing, e.g., that there is in general no relationship between intrinsic dimension and curvature.

Reviewers arrived at a mixed assessment of the paper’s contribution. All reviewers felt the curvature of feature representations to be an important topic, worthy of study. At the same time, reviewers raised two main issues. The first regards insight into either the origins of the curvature phenomena, and its effect on learning: reviewers felt the work would be more impactful if it took steps to identify how the curvature gain at the final layer arises and how it helps generalization. The second main issue concerns the paper’s approach to neighborhood generation: the paper proposes to compute SVDs of images (as matrices) and modify the singular values of the images. Although the manuscript alludes to a motivation in CAML paper (Li 2018), this paper does not discuss SVD of images themselves. Reviewers also expressed a number of smaller concerns with the clarity of the paper’s presentation.

After substantive deliberation, the reviewers remained mixed in their assessment, with 3 of 4 reviewers recommending rejection. The paper addresses an important general theme (understanding the geometry of feature representations in deep networks). At the same time, the paper would benefit from revision to make the conclusions less dependent on on SVD-based sampling (focusing the paper more on image manifolds, as discussed in the authors’ responses seems a good step), and from the addition of some preliminary insights into how curvature aids classification. In current form it falls below the bar for acceptance.


**Summary Of Ac-Reviewer Meeting:**

The AC and all four reviewers met, and had a substantive discussion of the merits and issues with the paper.

We discussed the following issues:

* Strengths of the paper:
    - [S1] geometry of feature representations is an important issue, and curvature in particular is an interesting lens
    - [S2] the paper makes two interesting observations: a characteristic jump-flat-jump curvature graph, and correlations between normalized curvature gap and generalization performance

Reviewers all agreed on [S1], and expressed some degree of agreement on [S2], albeit with different evaluations of quality and significant.

* Issue 1: SVD based sampling and neighborhood generation
    - two reviewers expressed concern about what the experiments of the paper are measuring: curvature is computed using neighborhoods of points generated in the following way: the paper takes a given sample (image), treats it as a matrix, and manipulates the singular values to produce new images. Reviewers expressed concern with whether this procedure actually generates a neighborhood of points on the image manifold.
   - discussion clarified that the SVD based scheme is a choice of the paper, not the CAML paper of Li, whose curvature estimator is used here. This had been a point of confusion in the initial discussion, since the paper (and author response) presents this choice as justified by the Li paper, when in fact the Li paper uses the SVD in a different manner (on batches of samples, *not* on a single sample).
   - Reviewers discussed whether this is a true concern, since images generated by SVD look like natural images. It was pointed out that noisy versions of the input (with small noise) have the same property, highlighting that for a second order quantity such as curvature, it is differences-of-differences which need to be plausible, not just the images themselves.
   - the authors' responses handle this issue in two ways (i) by citing the Li paper and asserting that SVD is related to curvature, and (ii) by quoting experiments on image transformation manifolds. (i) is not fully convincing, since, as noted above, the Li paper uses the SVD in a different way. (ii) would make for a better paper, if more focus was placed on it.

* Issue 2: Mean average curvature and other curvature measures
   - this was raised by one reviewer as a concern, as the paper's experiments present an incomplete picture of manifold curvature. After discussion it was agreed that this concern is less significant, as the paper is a first step, and in any case, one needs a summary statistic to present graphs

* Issue 3: Level of insight
   - several reviewers raised concerns about the descriptive nature of the paper, and the insights it provides into the origins and implications of the curvature phenomenon.
   - opinion on this issue remained split: one reviewer argued strongly that the paper presents new phenomena, and so the lack of explanation should not be a significant demerit. Another reviewer argued that the level of explanation and evidence falls below the standard of other works of this type (e.g., cited works on dimensionality of feature representation).

* Minor issues: reviewers briefly discussed other issues of clarity, but found these to be less essential than the three issues discussed above.

Overall, the AC experienced this meeting as a very substantive exchange. Although reviewer opinions on Issues 1 and 3 did not change much, discussion clarified the nature of the issues.